# Identification of global inhibitors of cellular glycosylation

Daniel Madriz Sørensen [1,12], Christian Büll[1,2,12], Thomas D. Madsen [1,3], Erandi Lira-Navarrete[1,4,5], Thomas Mandel Clausen[1,6,7], Alex E. Clark [8], Aaron F. Garretson[8], Richard Karlsson [1], Johan F. A. Pijnenborg [9], Xin Yin[10], Rebecca L. Miller [1], Sumit K. Chanda[10], Thomas J. Boltje[9], Katrine T. Schjoldager [1], Sergey Y. Vakhrushev [1], Adnan Halim[1], Jeffrey D. Esko[7], Aaron F. Carlin [7], Ramon Hurtado-Guerrero [1,4,5], Roberto Weigert [3], Henrik Clausen [1] ✉ & Yoshiki Narimatsu [1,11] ✉

Small molecule inhibitors of glycosylation enzymes are valuable tools for dissecting glycan functions and potential drug candidates. Screening for inhibitors of glycosyltransferases are mainly performed by in vitro enzyme assays with difficulties moving candidates to cells and animals. Here, we circumvent this by employing a cell-based screening assay using glycoengineered cells expressing tailored reporter glycoproteins. We focused on GalNAc-type O-glycosylation and selected the GalNAc-T11 isoenzyme that selectively glycosylates endocytic low-density lipoprotein receptor (LDLR)-related proteins as targets. Our screen of a limited small molecule compound library did not identify selective inhibitors of GalNAc-T11, however, we identify two compounds that broadly inhibited Golgi-localized glycosylation processes. These compounds mediate the reversible fragmentation of the Golgi system without affecting secretion. We demonstrate how these inhibitors can be used to manipulate glycosylation in cells to induce expression of truncated O-glycans and augment binding of cancer-specific Tn-glycoprotein antibodies and to inhibit expression of heparan sulfate and binding and infection of SARS-CoV-2.

Small-molecule inhibitors of cellular glycosylation are highly useful to probe the involvement of glycans in biological interactions and functions, and they have potential therapeutic applications in diseases such as cancer, inflammation, and infectious diseases[1–3].

Several natural compounds and synthetic sugar mimetics are available that alter or inhibit the glycosylation pathways or specific glycosylation steps in mammalian cells[2]. Natural compounds such as tunicamycin (derived from *Streptomyces lysosuperificus*) and

[1]Copenhagen Center for Glycomics, Departments of Cellular and Molecular Medicine, Faculty of Health Sciences, University of Copenhagen, Blegdamsvej 3, Copenhagen, Denmark. [2]Department of Biomolecular Chemistry, Institute for Molecules and Materials, Radboud University, 6525 AJ Nijmegen, The Netherlands. [3]Laboratory of Cellular and Molecular Biology, Center for Cancer Research, National Cancer Institute, National Institutes of Health, Bethesda, MD, USA. [4]The Institute for Biocomputation and Physics of Complex Systems (BIFI), Mariano Esquillor s/n, Campus Rio Ebro, 50018 Zaragoza, Spain. [5]Fundación ARAID, 50018 Zaragoza, Spain. [6]John A. Burns School of Medicine, University of Hawaii, Honolulu, HI, USA. [7]Department of Cellular and Molecular Medicine, University of California, San Diego, La Jolla, CA 92093, USA. [8]Department of Medicine, University of California, San Diego, La Jolla, CA 92093, USA. [9]Institute for Molecules and Materials, Department of Synthetic Organic Chemistry, Radboud University Nijmegen, Heyendaalseweg 135, 6525 AJ Nijmegen, The Netherlands. [10]Immunity and Pathogenesis Program, Infectious and Inflammatory Disease Center, Sanford Burnham Prebys Medical Discovery Institute, 10901 North Torrey Pines Road, La Jolla, CA 92037, USA. [11]GlycoDisplay ApS, Copenhagen, Denmark. [12]These authors contributed equally: Daniel Madriz Sørensen, Christian Büll. ✉e-mail: hclau@sund.ku.dk; yoshiki@sund.ku.dk

swainsonine (derived from *Swainsona canescens*) are commonly used to interfere with N-linked glycosylation[4,5], and inhibitors for glycolipid[6–8], and glycosaminoglycan[9–12] biosynthesis have been developed. Many of these compounds inhibit glycosidases or glycosyltransferases involved in the initiation or the early biosynthetic steps of the respective glycosylation pathways with varying selectivity and off-target effects. Inhibitors such as Brefeldin A (BFA, derived from *Penicillium brefeldianum*) rapidly block protein transport from ER to Golgi by preventing COP-I assembly and thus broadly affect protein glycosylation by preventing access of nascent ER glycoproteins to the Golgi-located glycosylation processes[13–15]. Since BFA blocks secretion and transport of glycoproteins to the cell surface, such inhibitors are of limited use as global inhibitors of glycosylation and to study the biological roles of glycans. More selective pan-inhibitors of sialyltransferase[16,17] and fucosyltransferase[17,18] isoenzymes based on fluorinated sugar mimetics enable specific deletion of glycan capping.

One challenge is to identify selective inhibitors of individual glycosyltransferase (iso)enzymes active inside the cell that block their specific glycosylation of unique protein targets or distinct pools of glycoproteins. A prime target for inhibitors are the 20 polypeptide GalNAc-transferase (GalNAc-Ts) isoenzymes that transfer N-Acetylgalactosamine (GalNAc) from the nucleotide sugar donor UDP-GalNAc to the hydroxyl group of serine, threonine, and possibly tyrosine amino acids forming the GalNAcα1-O-Ser/Thr/Tyr linkage, a step that initiates the GalNAc-type protein O-glycosylation (hereafter simply O-glycosylation unless otherwise defined)[19–21]. O-glycans are found on more than 80% of proteins trafficking the secretory pathway, and O-glycosylation serves important roles in fine-tuning protein functions including co-regulating pro-protein processing, ectodomain shedding, signaling, receptor dimerization, and receptor function[19,22–25]. For example, site-specific O-glycosylation in the ligand-binding domains of members of the low-density lipoprotein receptor (LDLR) and LDLR-related proteins (LRPs) catalyzed by the GalNAc-T11 isoenzyme markedly enhance interactions with ligands, e.g., LDL and VLDL[26,27]. Moreover, aberrant expression of immature O-glycans and overexpression of densely glycosylated carrier proteins, such as mucins, are characteristic features observed in many forms of cancer and are thought to contribute to cancer progression[28,29]. Inhibitors of O-glycosylation and in particular GalNAc-T isoenzymes that regulate the initiation step are still poorly explored and only few inhibitor candidates have been reported[30–34]. Most small-molecule inhibitors of O-glycosylation have been identified through in vitro enzyme assay screening of different GalNAc-Ts, and identified inhibitors include plant-derived flavonoids such as luteolin[35] and UDP-GalNAc-analogs[31,36]. However, most of these have faced challenges with selectivity including selectivity for particular GalNAc-T isoenzymes. A major limitation for discovery of selective inhibitors of glycosyltransferases suitable for in vivo studies is the lack of robust screening systems capable of discerning specificity of inhibitors for individual glycosyltransferases and in particular members of large isoenzyme families[2,37]. With facile gene-editing technologies it is now possible to engineer and custom design the glycosylation of mammalian cells for cell-based screening campaigns[38–40]. The first use of such glycoengineered cell models for inhibitor screens were employed by Linstedt and colleagues, using fluorescence biosensors for specific GalNAc-T isoenzymes to screen for isoenzyme-specific inhibitors[30,41]. This enabled the discovery of T3Inh-1 a selective inhibitor of GalNAc-T3 in vivo through the expression of these biosensors in isogenic cells with and without the relevant *GALNT*[30]. In a different example, selective inhibitors of STT3B were identified by using the genetically inactivated catalytic subunits of the OST, STT3A, and STT3B during screening. The identified inhibitors allowed for the regulation of N-glycan sites targeted by posttranslational N-glycosylation[42]. These examples show how glycoengineering of mammalian cell lines provides cell-based platforms that can be used

to screen for selective inhibitors of specific biosynthetic steps and isoenzymes.

Here, we used glycoengineering to develop cell-based assays for screening for inhibitors of one of the many GalNAc-T isoenzymes that initiate protein O-glycosylation (GalNAc-T11) and subsequently for general inhibitors of O-glycosylation. We screened a library of 1952 small-molecule compounds and identified two compounds, NSC80997 and NSC255112, which potently inhibit not only O-glycosylation, but also other Golgi-localized glycosylation processes, including elaboration of N-glycosylation and biosynthesis of glycosaminoglycans, notably without substantially affecting secretion of glycoproteins. Detailed studies into the mechanism of action for these compounds suggest that the effects are mediated through reversible fragmentation of the Golgi apparatus. We demonstrate that the identified compounds are widely useful for reversible inhibition of glycosylation in cell models without affecting glycoprotein transport and secretion, which opens widely up for studies of general roles of elaborate glycosylation features in cellular assays. The developed cell-based assays using secreted reporters were demonstrated to be suitable for screening of specific inhibitors of glycosyltransferases, and our discovery of global inhibitors of glycosylation in a small compound library stresses the need for thorough validation in further screening of larger compound libraries.

## Results

### A cell-based assay for high-throughput screening for inhibitors of O-glycosylation using secreted Tn-LDLR and T-MUC1 reporters

GalNAc-T11 specifically initiates O-glycosylation of the linkers between the ligand-binding class A (LA) repeats of members of the LDLR and related (LRP) receptor family at the $C_6XXX\underline{T}C_1$ position[22]. This GalNAc-T isoenzyme is not predicted to function with the biosensor strategy developed by Linstedt and colleagues[27,37] employing furin cleavable substrates given the selective substrate specificity for the LA repeats. To screen for potential selective inhibitors of GalNAc-T11, we therefore designed a reporter assay based on a secreted LDLR substrate (Fig. 1a, upper part). First, a secreted 6xHis-tagged chimeric CFP-LDLR reporter was designed for stable expression in HEK293 SimpleCells (SC) (HEK$^{KO\ COSMC}$), which endogenously express *GALNT11* and secrete the reporter with homogenous Tn O-glycans (GalNAcα1-O-Ser/Thr). Second, we developed an enzyme-linked lectin assay (ELLA) for quantitative capture of the secreted LDLR reporter in culture medium by anti-6xHis antibody and detection of Tn O-glycans introduced by GalNAc-T11 using the Tn-specific VVA lectin. Reporter secretion and capture efficiency was monitored by CFP fluorescence, and it was confirmed that the LDLR reporter was specifically O-glycosylated by GalNAc-T11 in wildtype HEK$^{WT}$ and HEK$^{KO\ COSMC}$ cells by demonstrating loss of O-glycans when expressed in HEK$^{KO\ GALNT11}$ cells with no differences in secretion levels (Supplementary Fig. 1A). The engineered HEK$^{KO\ COSMC}$ cells stably secreting the Tn-LDLR reporter were incubated with a compound library containing 1952 small molecules previously selected to have no/low cytotoxicity (National Cancer Institute). Cells were cultured in 96-well plates and treated for 24 hours (h) with 10 μM of the respective candidate compound and DMSO treatment was used as vehicle control. MTT assays of the treated cells revealed that ca. 30% of the compounds significantly altered metabolic activity, and those compounds were excluded from further analysis (Supplementary Fig. 1B). The culture medium collected after 24 h was analyzed with the capture ELLA to assess loss of Tn O-glycosylation of the secreted LDLR reporter. The first screening identified 45 compounds that reduced VVA binding (>60%) to the Tn-LDLR reporter compared to DMSO control (Fig. 1b). This number was reduced to two compounds, NSC80997 and NSC255112, after repeated screening and considering viability of cells and secretion levels of the reporter (Supplementary Fig. 1C). We then tested the effects of a subset of these identified

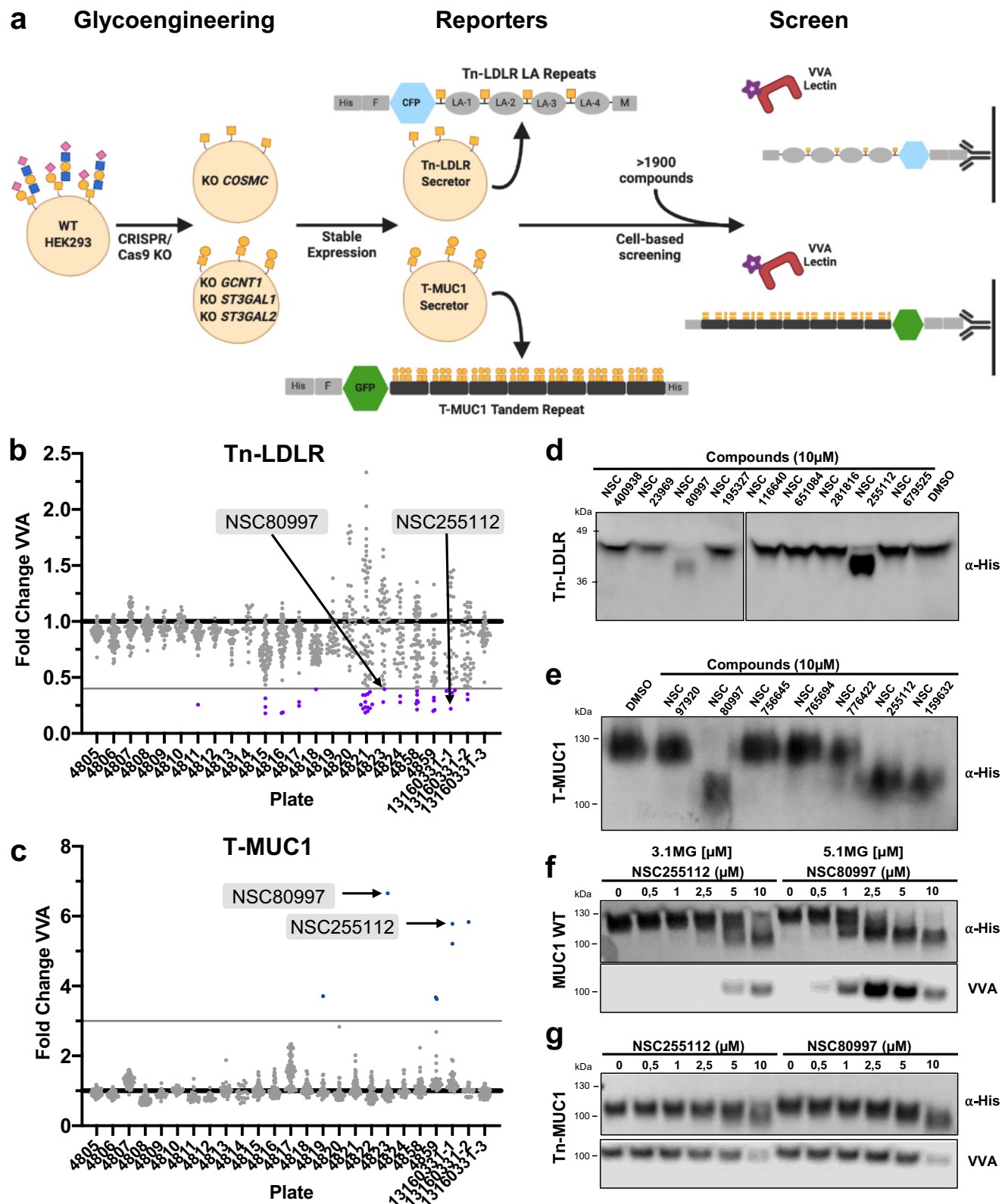

compounds with HEK[WT] cells (mainly producing mono- and disialyl T, and sialylated core2 O-glycan structures)[39] expressing the LDLR reporter. Sodium dodecyl-sulfate polyacrylamide gel electrophoresis (SDS-PAGE) and western blot analysis of the culture medium was used to evaluate potential changes in glycosylation and showed that only NSC80997 and NSC255112 robustly reduced the apparent molecular weight of the secreted LDLR reporter (Fig. 1d). These results suggested

that NSC80997 and NSC255112 interfered with O-glycosylation of the LDLR reporter.

We also developed a related cell-based assay platform suitable for screening of inhibitors of O-glycan elongation. HEK[WT] cells were engineered for core1 O-glycosylation without terminal α2-3 linked sialic acids (HEK293[KO GCNT1/ST3GAL1/2]) to enable detection of core1 by PNA lectin (T, Galβ1-3GalNAcα1-O-Ser/Thr), and the emergence of Tn

**Fig. 1 | Glycoengineered cell-based high-throughput screening identifies lead compounds altering O-glycan reporter protein glycosylation. a** Schematic presentation of the glycoengineering strategy yielding HEK293 (HEK) cells that stably secrete Tn-LDLR (HEK$^{KO\ COSMC}$) or T-MUC1 (HEK$^{KO\ GCNT1,\ ST3GAL1/2}$) reporter proteins, respectively, for cell-based HTP screening of 1952 compounds in a 96-well plate format by a VVA lectin ELLA (enzyme-linked lectin assay) detecting loss/gain of the Tn epitope on the reporter proteins. Secreted protein reporters include 6x-His-tag (His), FLAG-tag (F) and Myc-tag (M). **b, c** Dot plots show binding of VVA to LDLR (**b**) or MUC1 (**c**) reporter proteins secreted by glycoengineered cells treated with the 1952 compounds organized per 96-well plate, each containing 80 compounds and 3 DMSO vehicle controls. VVA binding is presented as fold-change binding normalized to DMSO control (black line). Active compounds were selected based on cut-off values (gray lines) for VVA binding to Tn-LDLR ( > 4-fold decrease) and to T-MUC1 ( > 3-fold increase). **c, e** Representative western blots show changes in molecular weight of the Tn-LDLR (**c**) and T-MUC1 (**d**) reporters secreted from HEK$^{WT}$ cells treated with 10 μM of each lead compound or DMSO control for 24 h. Reporters were detected by anti-6x His-tag antibodies. **f, g** Western blots show the MUC1 reporter produced in HEK$^{WT}$ (**f**) and HEK$^{KO\ COSMC}$ (**g**) cells incubated for 24 h with 0–10 μM NSC80997 or NSC255112 and detected by anti-6xHis-tag antibody (upper panel) and VVA (lower panel).

antigen by VVA upon inhibition of the core1 synthase C1GalT1 (Fig. 1a, lower part). For this, we designed a different reporter based on the tandem repeat region of the human mucin 1 (MUC1)[43], as this would also enable detection of general inhibitors of initiation of O-glycosylation (not limited to GalNAc-T11) by combined monitoring of loss of PNA and gain of VVA binding. The secreted T-MUC1 reporter was stably expressed in HEK$^{KO\ GCNT1/ST3GAL1/2}$ cells, and the compound library was tested as described for the LDLR reporter. The PNA detection assay produced low signal to noise ratio and it was not possible to reliably detect reduction in PNA signals. However, seven compounds were found to induce VVA reactivity with the reporter (Fig. 1c), surprisingly including the NSC80997 and NSC255112 compounds found to reduce VVA reactivity with the LDLR reporter expressed with Tn O-glycans (Fig. 1b). Testing the effects of the seven compounds with the MUC1 reporter expressed in HEK$^{WT}$ cells by Western blot analysis revealed that three of these induced marked changes in migration, including NSC80997 and NSC255112 (Fig. 1e).

These results suggested that NSC80997 and NSC255112 affected both initiation and elongation of O-glycosylation in ways not directly dependent on GalNAc-T11. We confirmed that neither compound directly targeted recombinant GalNAc-T11 by an in vitro glycosylation assay using non-glycosylated (naked) LDLR reporter protein produced in *E. coli* as substrate (Supplementary Fig. 1D). Further analysis by Western blot of the MUC1 reporter secreted by HEK$^{WT}$ and HEK$^{KO\ COSMC}$ cells revealed that both NSC80997 and NSC255112 reduced the molecular weight in a dose-dependent manner. Moreover, Tn antigen was detectable on the MUC1 reporter secreted from HEK$^{WT}$ cells treated with 2.5–10 μM of NSC255112 or 0.5–10 μM of NSC80997 compounds, with higher doses showing decreased VVA staining (Fig. 1f). Detection of Tn also decreased with concentrations >5 μM on the MUC1 reporter secreted from HEK$^{KO\ COSMC}$ cells (Fig. 1g). Taken together, this suggests a dose-dependent inhibition of O-glycosylation by NSC80997 and NSC255112, with lower doses inhibiting glycan elongation and consequently exposure of Tn, and higher doses further inhibiting the initiation of O-glycosylation. These results demonstrate the importance for careful evaluation of candidate compounds by different reporter designs for glycosylation, and indicate that future studies of larger compound libraries will require both assays for identification of inhibitors of specific isoenzymes.

## NSC80997 and NSC255112 induce Tn expression in human cell lines

The findings prompted us to assess the effects of NSC80997 and NSC255112 on O-glycosylation in different cell types. Incubation of HEK$^{WT}$ cells with increasing concentrations of both compounds (or equal volumes of DMSO as control) for 24 h showed that concentrations up to 10 μM were well-tolerated (Fig. 2a). Flow cytometry analysis of HEK$^{WT}$ cells treated with >1 μM revealed strong induction of VVA reactivity (Tn) on the surface, whereas DMSO control-treated cells remained VVA negative (Fig. 2b, c). The same effect was observed when probing with the 1E3 monoclonal antibody (mAb) that specifically recognizes the Tn antigen confirming display of the Tn antigen at the cell surface[44] (Fig. 2c). Tn induction was also observed in other human cancer cell lines including MCF-7, AGS, SH-SY5Y, and HeLa cells

(Fig. 2d). Kinetics studies demonstrated that Tn induction was detectable in HEK293$^{WT}$ and HeLa$^{WT}$ cells after 6–8 h incubation with a maximum being reached at 18 h (Fig. 2e). Notably, upon removal of the compounds from the culture, Tn expression was reduced more than half after 24 h of culture and lost around 72 h (Fig. 2f). Thus, both compounds readily induced Tn expression in HEK$^{WT}$ cells and this effect was reversible with no significant impact on cell viability or proliferation.

The biosynthetic capacities for initiation and core1 extension of O-glycans can be studied in the CHO ldlD mutant cell line that harbors a defect in the UDP-Gal 4′-epimerase (GALE) that converts UDP-Glc/GlcNAc to UDP-Gal/GalNAc[45,46]. The defect results in loss of UDP-Gal and all types of galactosylation including core1 synthesis and UDP-GalNAc and primarily initiation of O-glycosylation, but these effects can selectively be rescued by supplementation of exogenous Gal and/or GalNAc. We therefore tested if addition of Gal and/or GalNAc to HEK$^{WT}$ cells treated with NSC80997 and NSC255112 could rescue the inhibition of O-glycosylation and prevent Tn-induction (Supplementary Fig. 2), but no evidence of rescue was observed indicating that the compounds likely did not affect GALE and the epimerization of UDP-Glc/GlcNAc for donor substrate availability. Altogether these findings indicated that NSC80997 and NSC255112 induced Tn expression in a reversible manner, and that this effect is likely not mediated through interference with UDP-Gal/GalNAc biosynthesis.

## Remodeling of O-glycosylation by lead compounds is independent of the glucocorticoid receptor

NSC255112 is a derivate of geldanamycin, a known HSP90 inhibitor that is explored as an anti-tumor antibiotic[47,48]. NSC80997 is a derivate of cortivazol, a steroid that acts as high-affinity ligand for the glucocorticoid receptor (GR)[49]. Although, both NSC80997 and NSC255112 are structurally highly different (Fig. 3a), their known intracellular targets, HSP90 and GR, pointed to a potential common link with the glucocorticoid signaling pathway. HSP90 is a constitutively expressed chaperone that complexes with the GR and other chaperones in the cytoplasm, and upon steroid binding the GR translocates to the nucleus where it induces anti-inflammatory genes and represses pro-inflammatory genes[50,51]. We therefore examined if the observed effects of the compounds were mediated by the GR pathway. Using fluorescence microscopy, we found that treatment with NSC80997 and NSC255112 resulted in nuclear translocation of the GR in MCF7 cells at similar concentrations as the known GR agonist dexamethasone, suggesting that the effect on O-glycosylation could be GR-dependent (Fig. 3b and Supplementary Fig. 3A). However, a series of known GR agonists (dexamethasone, cortisone, cortisol, mometasone furoate) tested did not induce Tn expression, even at concentrations >100 μM (Fig. 3c and Supplementary Fig. 3B). Furthermore, pre-treatment of cells with mifepristone, a high-affinity GR antagonist, did not reduce NSC255112 or NSC80997-mediated induction of Tn on the MUC1 reporter (Supplementary Fig. 3c). Accordingly, induction of Tn was still observed in HEK cells after KO of GR (HEK$^{KO\ NR3C1}$) and in cells with KO of the related androgen receptor (HEK$^{KO\ NR3C4}$) (Fig. 3d). NSC80997 only differs with deacylcortivazol (DAC) by the lack of an α-hydroxyl group next to its ketone. This hydroxymethyl ketone in DAC forms an

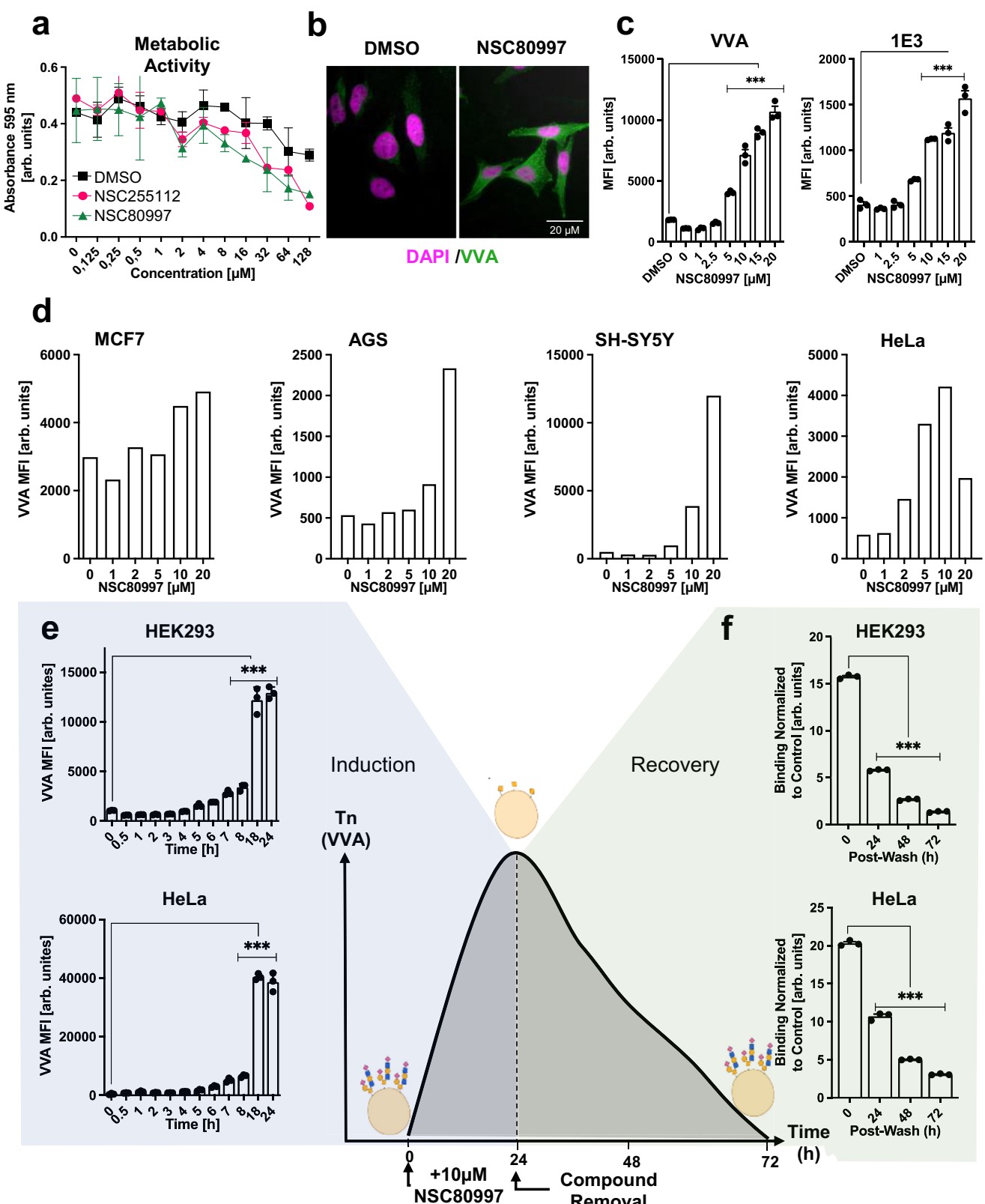

extensive network of hydrogen bonds with the GR receptor[49]. Reducing the methyl ketone to a secondary hydroxyl significantly decreased the ability of NSC80997 to induce Tn expression, suggesting that this part of the molecule could be involved in mediating the effects on glycosylation (Fig. 3e, f). Noteworthy, we observed a slight increase in VVA binding to HEK[WT] cells treated with other known HSP90 inhibitors including geldanamycin and chemical derivatives thereof, but these

compounds were highly cytotoxic in the nanomolar range (Supplementary Fig. 3D). We made multiple attempts to target the constitutively expressed HSP90 to further confirm this, but we were unable to obtain viable KO cells. In summary, our findings indicate that the effect of NSC80997 and NSC255112 on glycosylation is not directly mediated through the GR pathway, however, it is not inconceivable that inhibition of HSP90 could at least in part be involved.

**Fig. 2 | NSC80997 and NSC255112 induce Tn expression in human cell lines.**
**a** Line graph shows metabolic activity of HEK^WT cells treated with 0–128 μM NSC80997 and NSC255112 for 24 h measured by MTT assay. Data are presented as average 595 nm absorbance values ± SEM from three independent experiments.
**b** Representative images show HeLa^WT cells treated for 24 h with DMSO or 10 μM NSC80997 and stained with VVA lectin (green) and DAPI (magenta). Scale bar represents 20 μm. **c** Bar diagrams show cell surface Tn induction in HEK^WT cells treated with increasing concentrations of NSC80997 or DMSO control for 24 h. Binding of VVA (left) or anti-Tn (1E3) monoclonal antibody (right) was quantified by flow cytometry. Data from three independent experiments are presented as average mean fluorescence intensity (MFI) ± SEM (arbitrary unit) ***$P < 0.0001$ (ANOVA). **d** Bar diagrams show MFI values of cell surface VVA binding to MCF7, AGS, SH-SY5Y, and HeLa cells treated with 0-20 μM NSC80997 for 24 h assessed by

flow cytometry. Data is representative of two independent experiments. **e**, **f** Time-course analysis of cell surface Tn induction by NSC80997 (**e**) and recovery of glycosylation after removal of NSC80997 from culture medium (**f**) in HEK293^WT cells (top) and HeLa^WT cells (bottom). Cells were treated with 10 μM NSC80997 and VVA binding was assessed at indicated timepoints by flow cytometry. Data from three independent experiments are presented as average mean fluorescence intensity (MFI) ± SEM (arbitrary units). ***$P < 0.0001$ (ANOVA) (**e**). Loss of surface Tn structures in HEK^WT (top) and HeLa^WT (bottom) cells after treatment with NSC80997 for 24 h and extensive washing was detected by VVA staining and flow cytometry analysis 24, 48, and 72 h after treatment (**f**). Data from three independent experiments are presented as average mean fluorescence intensity (MFI) ± SEM (arbitrary units). ***$P < 0.0001$ (ANOVA).

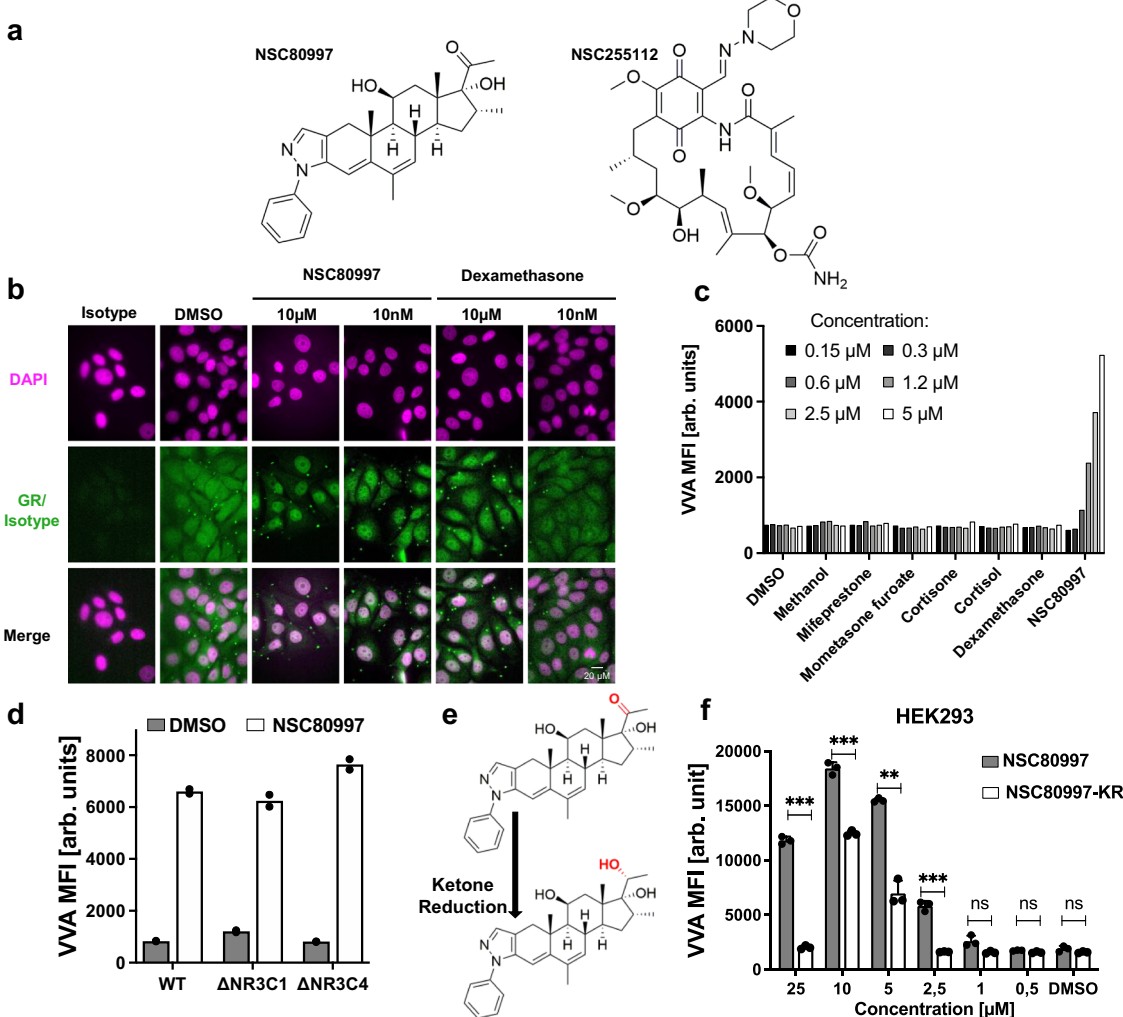

**Fig. 3 | Remodeling of glycosylation by lead compounds is independent of the glucocorticoid receptor (GR). a** Chemical structure of NSC80997 and NSC255112.
**b** Representative images of MCF7^WT cells treated with DMSO, 10 μM or 10 nm NSC80997 or dexamethasone for 24 h, stained with isotype or anti-GR antibody (green) and DAPI (magenta). Scale bar represents 20 μm. **c** Bar diagram shows cell surface VVA binding to HEK293^WT cells treated for 24 h with increasing concentrations of mifepristone, mometasone furoate, cortisone, cortisol, dexamethasone, or NSC80997. DMSO and methanol served as controls. Data was assessed by flow cytometry, is presented as mean fluorescence intensity (MFI) (arbitrary units) and is representative of two independent experiments.

**d** NSC80997-mediated Tn induction in GR knock-out (Δ) cell lines. Bar diagram shows VVA binding to HEK293^WT, HEK293^{KO NR3C1} and HEK293^{KO NR3C4} cells treated with 10 μM NSC80997 or DMSO control for 24 h. Lectin binding was assessed by flow cytometry and data are presented as average MFI values (arbitrary units) of two independent experiments. **e** The methyl ketone in NSC80997 was reduced to a secondary alcohol in NSC80997-KR highlighted in red. **f** Bar diagram shows VVA binding to HEK293^WT cells treated with increasing concentrations of NSC80997 or the ketone reduced derivative (NSC80997-KR). VVA binding was measured by flow cytometry and is presented as average MFI ± SEM from three independent experiments. ***$P < 0.0001$, **$P < 0,005$ (t-test).

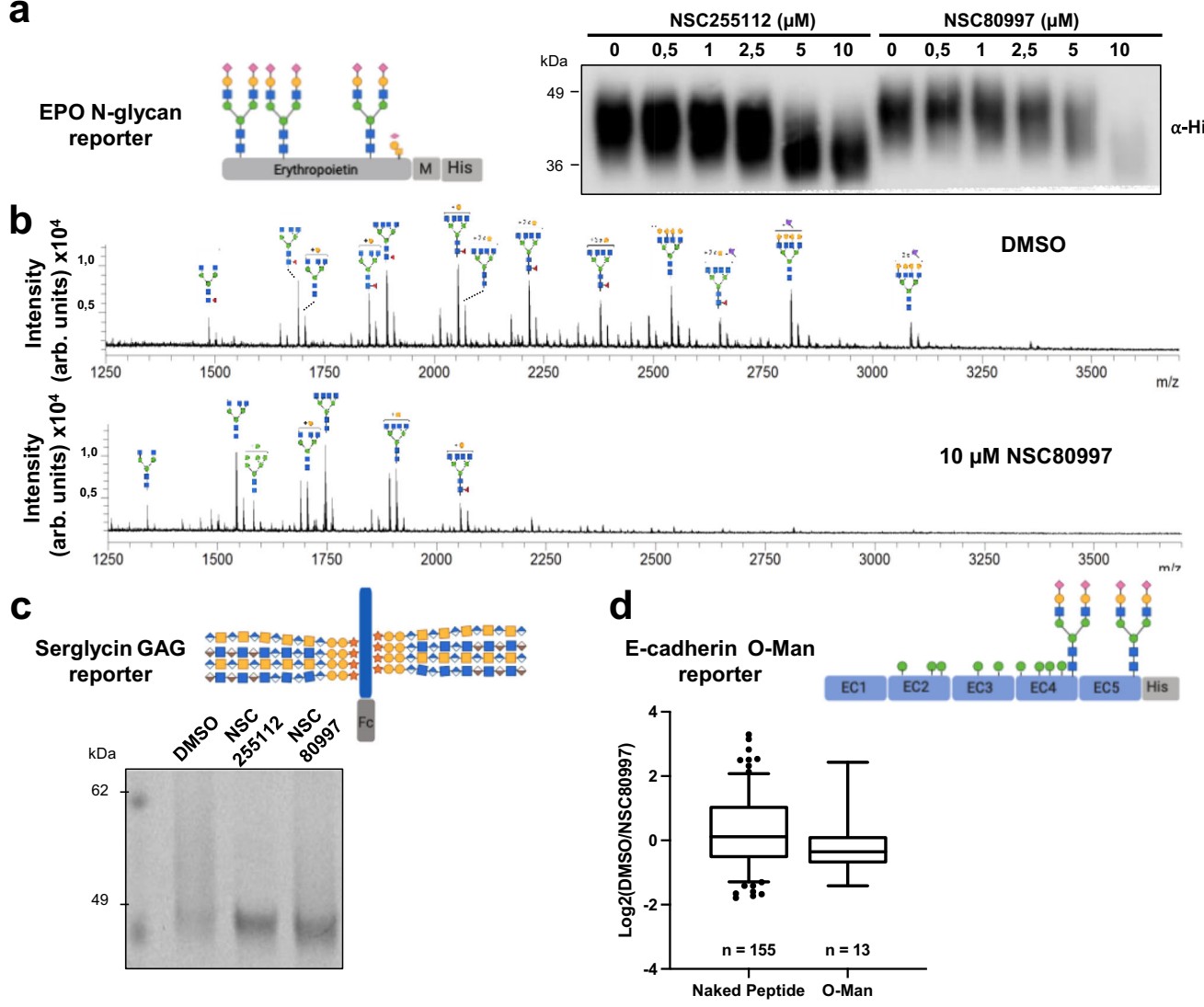

**Fig. 4 | NSC80997 and NSC255112 globally remodel cellular glycosylation.**
**a**, **b** N-glycan analysis of an erythropoietin (EPO) reporter by western blot and
MALDI-TOF analysis. **a** Western blot analysis of erythropoietin N-glycosylation
reporter secreted by CHO[WT] cells treated 24 h with 0–10 μM NSC80997 or
NSC255112 detected by staining with anti-6xHis-tag antibody. **b** MALDI-TOF pro-
filing of PNGaseF released N-glycans from erythropoietin reporter secreted by
HEK[WT] cells treated with 10 μM NSC80997 or DMSO control. **c** Coomassie gel
staining analysis of a glycosaminoglycan (GAG) serglycin reporter protein secreted

by CHO[WT] cells treated for 24 h with 10 μM of NSC80997, NSC255112 or DMSO
control. **d** Proteomic analysis of E-cadherin tryptic digest purified from DMSO
(TMT-129C channel) or NSC80997 (TMT-130N channel) treated HEK[WT] cells. Box
and whisker plot depicts Log2(DMSO/NSC80997) ratio distributions of identified
PSMS from non-glycosylated (naked) peptides and O-Man glycopeptides. Box plots
include a center line (median), box limits (upper and lower quartiles), whiskers
(10th–90th percentile). Source data is provided as a source file. Glycans are drawn
according to the SNFG nomenclature.

## NSC80997 and NSC255112 broadly remodel the cellular glycome

Next, we assessed whether the lead compounds affected glycosylation
pathways other than GalNAc-type O-glycosylation. We profiled treated
HEK[WT] cells by flow cytometry with a panel of lectins and found that
lectins recognizing α2-3/6-sialylation (Pan-Lectenz, α2,3-Lectenz),
agalactosylated complex N-glycans (GSLII), and high-mannose N-gly-
cans (GNA) were affected by both NSC80997 and NSC255112, sug-
gesting these compounds broadly alter cellular glycosylation
(Supplementary Fig. 4A). This prompted us to perform a more detailed
analysis using representative recombinant secreted proteins for dif-
ferent types of glycosylation. First, we investigated the effects on
N-glycosylation using erythropoietin (EPO) expressed in CHO cells,
and both NSC80997 and NSC255112 induced a clear shift in migration
of EPO at similar concentrations as the MUC1 reporter (Fig. 4a), and
profiling released N-glycans from purified EPO revealed a clear shift to
N-glycans with a marked reduction in galactosylation and almost

complete loss of sialylation upon NSC80997 treatment (Fig. 4b).
Interestingly, there were no major effects on the assembly of the basic
tetra-antennary N-glycan cores structures with full incorporation of
GlcNAc residues, although NSC80997 treatment seemed to also
increase the level of high-mannose N-glycans slightly. The results were
in line with the observed changes detected by lectin staining (Sup-
plementary Fig. 4A).

Next, we tested the effects on glycosaminoglycan (GAG) bio-
synthesis using recombinant expression of serglycin in CHO cells
(Fig. 4c), and treatment with both NSC80997 and NSC255112 resulted
in accumulation of the core protein without the characteristic high
molecular weight smear in SDS-PAGE analysis observed with treat-
ment, suggesting absence of GAG chains[52]. Finally, we investigated the
TMTC1-4 directed O-mannose type of protein glycosylation[53] using
recombinant expression of a secreted E-cadherin protein expressed in
HEK[WT] cells by mass spectrometry, and all eight potential O-Man

glycosites in the extracellular cadherin (EC) domains EC2-EC4 were glycosylated with and without treatment with NSC80997 (Fig. 4d).

In summary, these findings suggest that the identified compounds broadly interfere with several types of protein glycosylation including GalNAc-type O-glycosylation, N-glycosylation, and GAG biosynthesis, while TMTC driven O-mannosylation was unaffected. The TMTC1-4 directing O-mannosylation of the cadherin superfamily occurs in the ER[54,55], and all N-glycosylation steps located in the ER were also largely unaffected. Thus, the two compounds selectively affect glycosylation steps that occur in the Golgi, and interestingly mainly the steps involved in general elaboration of glycan structures and not the early N-glycan branching steps orchestrated by the MGATs[56] and to a lesser extend the initiation step of GalNAc-type O-glycosylation orchestrated by the GalNAc-T isoenzymes[21] that are both believed to occur from the earliest cis-Golgi stacks. Importantly, the effects on glycosylation were characterized with secreted reporter glycoproteins as well as lectin/antibody labeling of non-permeabilized cells demonstrating that the compounds did not affect the general transport and secretion of proteins.

## NSC80997 and NSC255112 trigger reversible fragmentation of the Golgi apparatus

Since both identified compounds impaired glycosylation steps in the Golgi apparatus, we tested their effect on the Golgi structure in HeLa cells. Treatment with NSC80997 induced fragmentation of the Golgi apparatus, as visualized by indirect immunofluorescence using antibodies directed against the cis/medial-Golgi marker giantin (Fig. 5a). A similar phenotype was observed 24 h after treatment with NSC255112 (Fig. 5c). The structure of the trans-Golgi area was also affected, as shown by probing sialylation in the medial/trans-Golgi system using alkyne-tagged sialic acid (Ac5SiaNPoc)[57] by click chemistry and staining with antibodies directed against the trans-Golgi marker TGN46 (Supplementary Fig. 5A, B). The effects of both compounds were reversible, as their removal from the culture medium resulted in re-assembly of the Golgi within 24 h (Fig. 5b, c and Supplementary Fig. 5B). Treatment with NSC80997 resulted in a loss of the staining for the Golgi-resident glycosyltransferase GalNAc-T1 (UH3, 4D8)[58] (Supplementary Fig. 5A) without affecting the total protein level of another Golgi glycosyltransferase GalNAc-T2 as evaluated by western blot analysis of total cell lysates (Supplementary Fig. 5C). This suggests that the loss of localized Golgi staining of glycosyltransferase proteins is due to redistribution rather than loss of protein in local concentrations below the detection limit of immunocytology with the used mAbs. Accordingly, proteomics analysis of HEK[WT] cells treated for 24 h with 10 μM NSC80997 showed limited changes in the proteome compared to DMSO-treated control cells (Supplementary Fig. 5D).

Depletion of HSP90 has previously linked to fragmentation of the Golgi by decreased levels of the microtubule-associated protein 4 (MAP4), which is important for microtubule acetylation and stabilization[59,60]. Given the hypothesis that effects of the identified compounds could be mediated through HSP90, we tested microtubule acetylation in HeLa cells after treatment with NSC80997 and observed no change in the level of acetylated or total tubulin in treated cells (Supplementary Fig. 5E). Similarly, we observed no effect on the NSC80997-mediated induction of cell surface Tn expression by pre-treatment or co-treatment with Trichostatin-A (TSA), which has been reported to promote microtubule acetylation[60,61] (Supplementary Fig. 5F). Thus, while a compelling target, these results suggest that the effects of the inhibitors are not the result of decreased microtubule acetylation mediated through HSP90.

## NSC80997 and NSC255112 function differently than Brefeldin A

To gain further insight into the mechanism of action of NSC80997, cells were transiently transfected with mScarlet-Giantin, and imaged using spinning-disk confocal microscopy. Treatment with NSC80997 revealed that the Golgi fragmentation was preceded by substantial degree of tubulation, which in principle could be reminiscent of the well-established effect of BFA (Fig. 5a and Supplementary Fig. 6A)[14,62]. However, further analysis of the changes in Golgi-morphology over time showed that while BFA induces the redistribution of the Golgi apparatus into the ER, NSC80997 only leads to dispersion of the Golgi markers (Fig. 5d, Supplementary Fig. 6A, Supplementary Movies 1–3), clearly indicating a different mechanism of action. This conclusion is further supported by three lines of additional evidence. First, βCOP, a downstream target of BFA, did not dissociate from the Golgi apparatus upon treatment with NSC80997[14] (Fig. 5e and Supplementary Fig. 6B). Second, BFA treatment completely abrogated secretion of the MUC1 reporter from HEK[WT] cells at 0.5 μM, whereas NSC80997 clearly did not, in agreement with the cell-based screening strategy relying on secreted reporters (Supplementary Fig. 6C). Finally, BFA treatment did not lead to induction of Tn antigen expression on the cell surface at any concentration tested, while NSC80997 strongly induced Tn cell surface expression starting from 1 μM (Supplementary Fig. 6D).

## NSC80997 and NSC255112 do not affect transport and secretion of glycoproteins

BFA is known to rapidly affect the Golgi and block transport of glycoproteins out of the ER (Fig. 5d), while induction of surface expression of Tn by NSC80997 and NSC255112 was detectable after 6 h and peaking at 18–24 h (Fig. 2e). It is challenging to study secretion of reporter glycoproteins in time-course experiments given the inherent delay in turnover of glycoproteins and that sufficient time is needed to produce detectable signal. In the experiments presented in Figs. 1–4 and Supplementary Figs. 1–3 the analyses of secreted glycoprotein reporters were based on total accumulated material 24 h after start of drug treatments. To better distinguish the effects of the drug treatments on secretion and glycosylation of reporters we introduced a wash-out 4 h after first exposure to the drugs followed by 24 h continued incubation in the presence of the compounds (Supplementary Fig. 6C). This demonstrated that NSC80997 at low concentrations markedly affected glycosylation at 0.5 μM, while the previous analysis without the wash-out appeared to indicate comparative effects only at 5 μM (Fig. 1f, e), due to accumulation of secreted reporters before and after the effect of NSC80997. The comparative analysis of BFA clearly confirms the effect of the wash-out and shows that BFA at 0.5 μM completely blocks secretion (Supplementary Fig. 6C). Clearly, the same challenge applies to changes of cell surface glycoproteins, where slow turnover of membrane glycoproteins adds to the complexity and extended time of treatment at lower concentrations of inhibitor compounds could be used for optimal effects without affecting cell viability (Supplementary Fig. 6D, E). In conclusion, the NSC80997 and NSC255112 compounds are unique tools that can be used at ranges of concentrations (0.5-5 μM) to block elaborated glycosylation features on cell surface and secreted glycoproteins.

## Induction of aberrant truncated cancer-associated O-glycans by NSC80997 enhances ADCC

Aberrant O-glycosylation such as expression of the truncated Tn antigen is commonly observed in different cancer types[29,63], and aberrant Tn O-glycopeptide epitopes can elicit high-affinity mAbs[64–66]. Thus, Tn-MUC1 is a promising target currently explored in cancer immunotherapies by mAbs or CAR-T cells with high affinity for this mucin glycoform[66–68]. We examined the induction of Tn-MUC1 by NSC80997 in cells stably expressing membrane anchored MUC1 reporter (HEK[MUC1]). Tn-MUC1 was induced and detected with mAb 5E5 directed against Tn-MUC1[65,68] (Fig. 6a). In parallel, we also tested a type I membrane protein FXYD5, known as dysadherin endogenously expressed by HEK[WT] cells, with a recently established Tn-FXYD5 mAb (6C5)[64] and a mAb directed to the FXYD5 protein (NCC-M53)[69]. Expression of the Tn-MUC1 epitope was strongly induced at

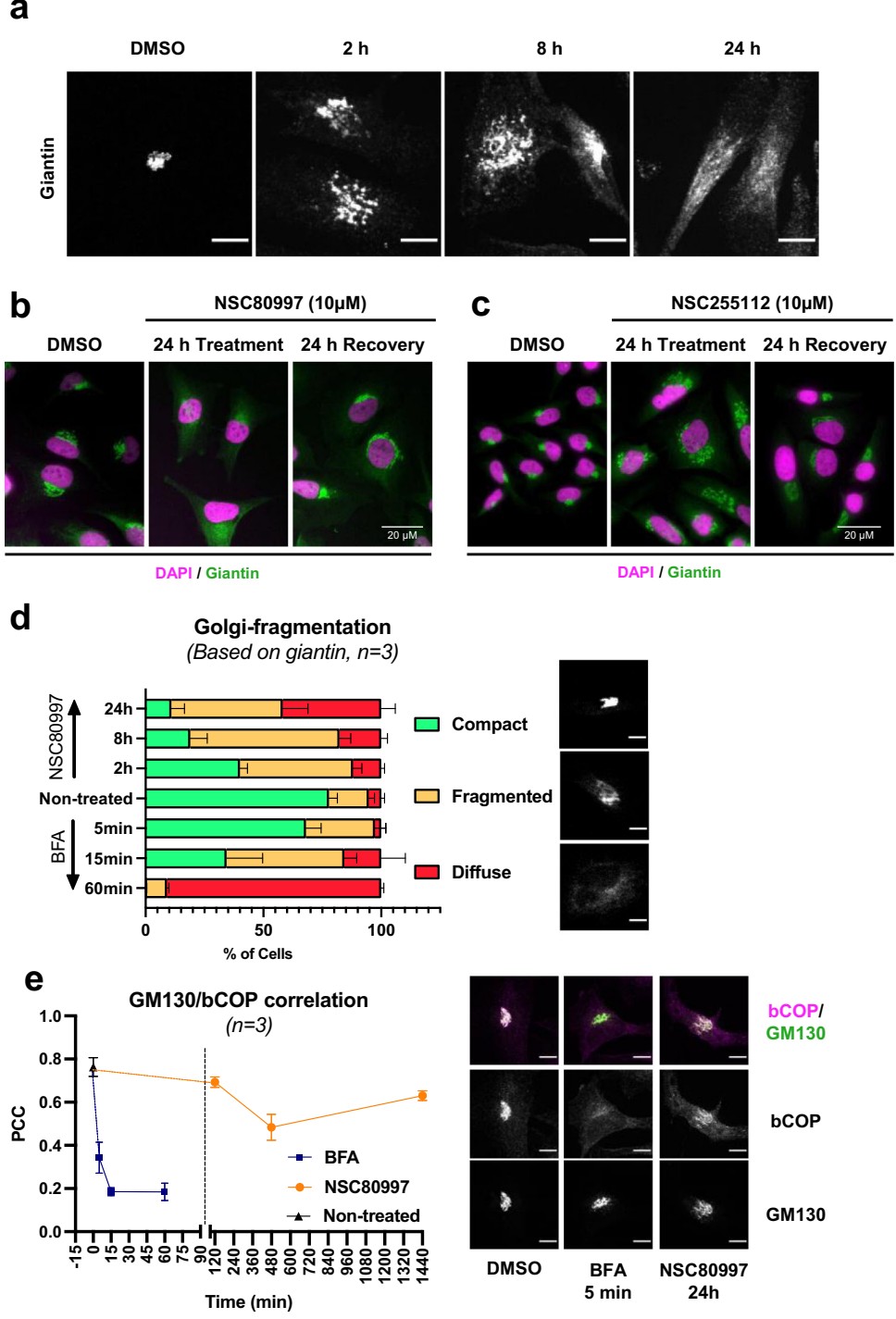

**Fig. 5 | NSC80997 and NSC255112 trigger reversible Golgi fragmentation.**
HeLa[WT] cells were treated with 10 μM NSC80997, NSC255112 or Brefeldin A and processed for immunofluorescence. **a**–**c** HeLa cells were imaged by confocal microscopy to reveal the morphology of the Golgi apparatus (giantin, green) and the nuclei (DAPI, magenta). **a** Time course of Golgi fragmentation during treatment with NSC80997. Scale bars represent 10 μm. **b**, **c** After treatment for 24 h with either NSC80997 (**b**) or NSC255112 (**c**), cells were washed with fresh medium and incubated for another 24 hrs without inhibitor. Scale bar represents 20 μm. **d** Bar plot shows percentage of HeLa[WT] cells displaying compact, fragmented or diffuse Golgi morphologies at different timepoints during treatment with

NSC80997. Data is presented as mean + SEM from three independent experiments with 62–109 cells/experiment. Example images depicting compact, fragmented or diffuse Golgi morphologies are shown. Scale bars represent 10 μm. **e** Line graph depicts Pearson's correlation coefficient of bCOP (magenta) versus cis-Golgi marker GM130 (green) at different timepoints during treatment with NSC80997 or Brefeldin A. Images were acquired as Z-stacks using spinning-disk microscopy and data is presented as mean ± SEM from three independent experiments with 10-15 cells/experiment. Representative images of co-localization are shown. Scale bars represent 10 μm.

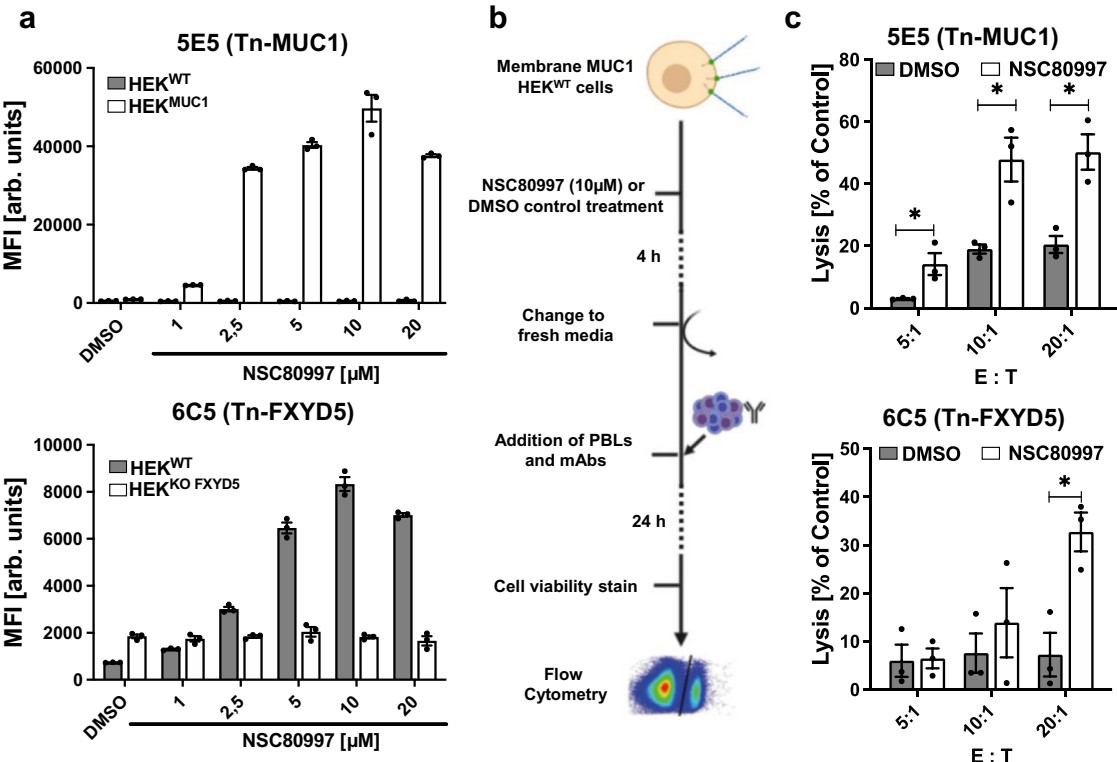

**Fig. 6 | NSC80997 boosts antibody-dependent cellular cytotoxicity (ADCC) of Tn-specific mAbs 6C5 and 5E5. a** Cell surface binding of mAbs 5E5 (top) and 6C5 (bottom) to HEK$^{WT}$, HEK$^{KO\ FXYD5}$ cells, and HEK$^{WT}$ cells stably expressing membrane MUC1 reporter protein (HEK$^{MUC1}$) treated with increasing concentrations of NSC80997 for 24 h. Antibody binding was measured by flow cytometry and data is shown as mean MFI ± SEM (arbitrary units) of three independent experiments. **b** Schematic illustration of ADCC assay. HEK$^{MUC1}$ cells were treated with 10 μM NSC80997 or DMSO for 4 h, washed, and re-seeded. Monoclonal antibodies directed against Tn-MUC1 (5E5) and Tn-FXYD5 (6C5) were added to the HEK cells 20 min prior to addition of CFSE-labeled peripheral blood lymphocytes (PBLs) at 5:1, 10:1, and 20:1 effector to target (E:T) ratios. Isotype (mouse IgG) treated cells served as control. After overnight incubation, all cells were harvested and labeled with viability dye and analyzed by flow cytometry. **c** Bar diagrams show percentage lysis normalized to isotype control, of DMSO or NSC80997 treated HEK$^{MUC1}$ cells induced by 5E5 (top) or HEK$^{WT}$ cells induced by 6C5 (bottom) mAbs. Data is presented as mean percentage lysis compared to control ± SEM from 3 independent experiments. *$P < 0.05$ (t-test).

compound concentrations >1 μM in the HEK$^{MUC1}$ cells, while no 5E5 binding was detected with treated HEK$^{WT}$ (Fig. 6a). Similarly, NSC80997 induced 6C5 binding to HEK293$^{WT}$ cells, but not to HEK293$^{KO\ FXYD5}$ cells, while NCC-M53 binding remained unaltered (Fig. 6a and Supplementary Fig. 7A). Noteworthy, pulsing the cells with 10 μM NSC80997 for 1–4 h was sufficient to obtain high Tn expression 24 h later (Supplementary Fig. 7B). We then used HEK$^{MUC1}$ cells pulsed for 4 h with 10 μM NSC80997 or DMSO control and performed an antibody-dependent cellular cytotoxicity (ADCC) assay. HEK$^{MUC1}$ target cells were co-cultured with different ratios of peripheral blood lymphocytes (PBLs) from healthy donors in the presence of 5E5 or isotype control antibody (Fig. 6b). Pre-treatment of the HEK$^{MUC1}$ target cells with NSC80997 significantly triggered ADCC for 5E5 and 6C5 mAbs increasing with PBLs ratios, but not with an irrelevant isotype matched mAb (Fig. 6c). Similar results were observed for the general Tn hapten specific mAb 1E3 (Supplementary Fig. 7C). These findings suggest that NSC80997 may be useful for boosting Tn antigen expression in cancer cells to enhance efficacy of immunotherapies targeting Tn glycoproteins.

### Reduction of cell surface heparan sulfate by NSC80997 treatment inhibits binding and entry of SARS-CoV-2

Widespread SARS-CoV-2 infection is currently a major health challenge worldwide, causing serious clinical morbidity and mortality[70]. Recent studies have provided evidence that cellular heparan sulfate (HS) is a necessary co-factor for SARS-CoV-2 infection by mediating binding of the receptor-binding domain (RBD) of the spike glycoprotein and the

human angiotensin-converting enzyme 2 (ACE2), and enzymatic degradation of HS blocks the spike protein binding and viral infection[71]. Since NSC80997 treatment reduced GAG formation on recombinant expressed Serglycin (Fig. 4c), we tested the potential effects of this compound on SARS-CoV-2 cell binding and virus entry. We first confirmed that NSC80997 blocked GAG biosynthesis by disaccharide HPLC analysis of isolated GAGs from total cell lysates of HEK$^{WT}$ cells treated with 10 μM NSC80997 for 24 h. We observed a clear reduction in the amount of chondroitin/dermatan sulfate (CS/DS) (Supplementary Fig. 8). Next, we found that treatment of Vero E6 and A549 cells with >5 μM of NSC80997 reduced surface binding of recombinant SARS-CoV-2 spike protein to similar levels as found with treatment by heparinase (Fig. 7a). Moreover, we assessed the effects of NSC80997 treatment in an in vitro infection model based on a pseudotyped vesicular stomatitis virus (VSV) expressing the full-length SARS-CoV-2 S protein that induces luciferase expression upon infection of Vero E6 cells[71,72]. Robust luciferase induction was observed in control-treated Vero E6 cells upon infection, and treatment with 10 μM of NSC80997 almost completely abrogated this (Fig. 7b). Lastly, we tested authentic SARS-CoV-2 virus infection using strain USA-WA1/2020. Infection of Vero-TMPRSS2 (Vero E6 cells stably expressing the transmembrane serine protease TMPRSS2), Caco-2, and Huh-7.5 cells was monitored by staining of the cells with antibodies against the SARS-CoV-2 nucleocapsid (N) protein. We observed robust reduction in viral infection upon treatment with increasing concentrations of NSC80997 in all cell lines, with IC50 values ranging from 4.3 μM for Caco-2 to 1.4 μM for Vero-TMPRSS2 and Huh-7.5 cells (Fig. 7c).

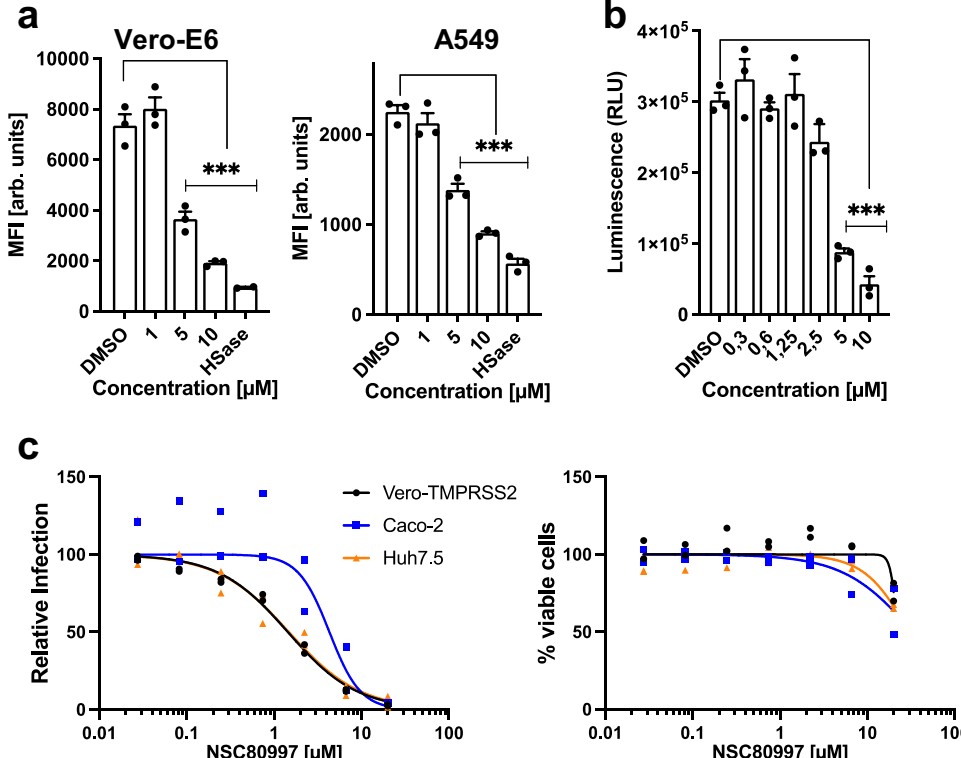

**Fig. 7 | Reduction of cellular heparan sulfate by NSC80997 treatment inhibits binding and entry of SARS-CoV-2. a** Cell surface binding of recombinant SARS-CoV-2 spike protein to Vero-E6 (left) and A549 (right) cells treated with increasing concentrations of NSC80997 or DMSO control for 24 h or after heparinase treatment (HSase). Binding was assessed by flow cytometry and is presented as mean fluorescence intensity (MFI) ± SEM (arbitrary units) from 3 independent experiments. ***$P < 0.0001$ (ANOVA). **b** Infection of NSC80997 or control-treated Vero-E6 cells with SARS-CoV-2 S protein pseudotyped virus expressing luciferase. Infection was measured by the addition of Bright-Glo and detection of luminescence. Data is presented as mean relative light units (RLU) ± SEM from 3 independent

experiments. ***$P < 0.0001$ (ANOVA). **c** Relative infection of SARS-CoV-2 (strain USA-WA1/2020) (left) and percent viable cells (right) in Vero-TMPRSS2, Caco-2, and Huh7.5 cells treated with increasing concentrations of NSC80997 or DMSO control. Infection was monitored by immunofluorescence using antibodies against SARS-CoV-2 nucleocapsid protein after 24 h pre-treatment with NSC80997 and 18 h exposure to virus with continued NSC80997 treatment. Cell viability was measured by Sytox Green nuclear stain. The experiments were performed in duplicate twice. The average values from each experiment were used to calculate the IC50 values and are shown as relative infection and cell counts compared to control, respectively. Non-linear fit curve is shown.

No decrease in cell viability was observed in cells treated with concentrations <10 μM NSC80997. These findings suggest that NSC80997-mediated inhibition of cellular HS biosynthesis reduces binding and entry of SARS-CoV-2 in this in vitro model.

## Discussion

In this study, we employed glycoengineered cells with stable secretion of tailored glycoprotein reporters for robust screening of a small-molecule library and discovery of inhibitors of glycosylation. We designed and employed two cell-based assay designs for discovery of selective inhibitors of the GalNAc-T11 (initiation of O-glycosylation) and C1GalT1 (core1 extension of O-glycosylation) glycosyltransferases. We tested a relatively small available compound library that has been optimized for cell-based screening and identified two distinct molecules serving as broad inhibitors of Golgi-located glycosylation processes. We did not identify selective inhibitors of the GalNAc-T11 and C1GalT1 enzymes within the limited compound library, but we validated the assay design with independent testing of consecutive biosynthetic steps, and further testing of larger compound libraries is clearly warranted. We did, however, identify two potent compounds, NSC80997 and NSC255112, that broadly inhibited glycosylation steps located to the Golgi including those involved in elaborating O-glycans, N-glycans and glycosaminoglycans. The NSC80997 and NSC255112 compounds induced reversible Golgi fragmentation within a few hours of treatment, but they did not cause redistribution of Golgi components into the ER and most importantly they did not affect transport

and secretion of glycoproteins as found with the commonly used Golgi-disrupting agent BFA. These inhibitors of Golgi glycosylation provide new tools to study the role of elaborated glycosylation features in cells at the cell surface and in secretion.

A major limitation for discovery of selective inhibitors of glycosyltransferases suitable for in vivo studies is the lack of robust screening systems capable of discerning specificity and cross-reactivity of inhibitors for individual glycosyltransferases and in particular members of large isoenzyme families[2,37]. Linstedt and colleagues[27,37] solved this challenge with bioassays reporting the cellular functions of several GalNAc-T isoenzymes, but this design is not universally applicable to all GalNAc-T isoenzymes or glycosyltransferases in general. We have developed engineered cell libraries with combinatorial KO/KI of glycosyltransferases in an array-able format to dissect the biosynthesis and genetic regulation of all types of cellular glycosylation[38–40,43,52,73–75], and these cell-based arrays provide screening platforms for glycosylation inhibitors. The engineered cells enable production of secreted glycoprotein reporters with custom-designed glycans that can inform of how inhibitors affect highly specific steps in protein glycosylation. Here, we originally aimed to demonstrate the potential of this platform by developing a cell assay to probe for specific inhibitors of the GalNAc-T11 isoform (Fig. 1a, upper part), and this illustrated the challenges with cell assays and highlighted the necessity to co-screen for broader effects on glycosylation. To meet the latter, we developed a cell assay that broadly probes for GalNAc-T isoenzymes using parts of the MUC1 mucin substrate (Fig. 1a,

lower part). Intriguingly, we identified two lead compounds that broadly abrogate elaboration of glycans in the Golgi apparatus without affecting protein secretion. The compound library was limited in size and future screening with larger compound libraries may lead to more selective inhibitors. However, the study clearly illustrates the potential of the cell-based array platform to evaluate and dissect effects of inhibitors on glycosyltransferase isoenzymes and individual biosynthetic steps in different glycosylation pathways.

The two compounds we did identify within the limited library, NSC80997 and NSC255112, however, turned out to have interesting properties with wide applicability as reversible inhibitors of elaborate glycan features such as galactosylation and sialylation associated with different types of glycoproteins and other glycoconjugates. Although structurally different, we found that both compounds readily and reversibly abrogated elongation of GalNAc-type O-glycosylation, processing of complex type N-glycans after the key branching step, and the biosynthesis of GAG chains. The biosynthetic steps affected are all predicted to occur in the Golgi, while glycosylation located in ER and early Golgi was unaffected. Treatment of HEK cells with these compounds did not affect secretion and did not appear to induce significant changes in the overall proteome, but immunocytology (and metabolic labeling of sialylation) demonstrated that Golgi-resident proteins dispersed within 1-2 h without being lost (Fig. 5a and Supplementary Fig. 5A, B). These effects were reversible upon removal of inhibitors, and the induction/recovery kinetics of the Golgi fragmentation and effects on reporter and surface glycosylation were highly similar, suggesting that the Golgi disruption directly led to the effects observed on protein glycosylation. Aberrant glycosylation is a common feature found in cells with fragmented Golgi apparatus including in cancer cells[76–79].

GalNAc-Ts are generally found throughout the Golgi stacks[80] and initiate O-glycosylation after the protein folding in ER. However, it should be noted that relocation of GalNAc-Ts to ER has been reported as a feature of cancer cells and cause of the characteristic expression of truncated Tn O-glycans in cancer[81]. We found that lower doses of the inhibitors abrogated O-glycan elongation but not the initiation leading to accumulation of Tn O-glycans (Fig. 1e) could potentially suggest that the inhibitors partly redistributed GalNAc-Ts to ER although this was not apparent from the immunocytology results. However, given that the N-glycan branching step by the MGATs was not affected substantially, it is more likely that the inhibitors have lower effects on glycosylation steps performed in the early *cis*-Golgi stacks.

NSC80997 is a derivate of cortivazol, a synthetic steroid that acts as high-affinity ligand for the GR[82–84], and these share the four ring (A-D) steroid structure and a phenylpyrazol group at the A-ring (Fig. 3a). This bulky group has been shown to induce structural changes in the GR ligand-binding pocket creating additional interactions between cortivazol and the GR and enhancing the binding affinity 20–50 times compared to dexamethasone[49,85,86]. Because of this increased binding affinity, cortivazol was assessed in clinical trials for corticosteroid-mediated pain relief after intra-articular, periarticular, or epidural injection without clear outcome or clear adverse effects, and apparently the use and production was terminated in 2017[87,88]. NSC255112 is a derivate of the HSP90 inhibitor geldanamycin. HSP90 and other proteins including HSP70 and FKBP4 interact with GR and retain GR in the cytoplasm[51,89]. Binding of a ligand releases the GR from HSP90 and allows for traffic to the nucleus and transcriptional regulation of inflammatory genes[51,90]. Binding of NSC80997 to the GR or inhibition of HSP90 by NSC255112 thus could potentially release the GR. We did observe shuttling of the GR to the nucleus after treatment with high and low concentrations of both compounds (Fig. 3B and Supplementary Fig. 3A), however, other GR steroid ligands tested did not show the same effects on glycosylation as NSC80997 and NSC255112. Moreover, KO of the GR gene did not abolish the effects of the inhibitor compounds, and treating cells

pretreated with the high-affinity GR antagonist mifepristone also did not affect the inhibition (Fig. 3c, d). These results indicate that NSC80997 and NSC255112 affect Golgi glycosylation largely independent of the GR pathway.

Notably, while the NSC255112 analog geldanamycin is cytotoxic, we noticed induction of Tn in cells treated with other known HSP90 inhibitors (Supplementary Fig. 3D), and inhibition and depletion of HSP90 have previously been linked to Golgi fragmentation[59,60]. HSP90 binds the microtubule-associated protein 4 (MAP4), which is essential for maintaining microtubule acetylation and stabilization. HSP90 depletion leads to decreased levels of MAP4 and thus fragmentation of the Golgi system, but effects on glycosylation have not been reported previously[60]. However, we observed no reduction in the levels of acetylated tubulin upon treatment with NSC80997, suggesting that the microtubule network remains intact. A study has proposed that loss of HSP90 leads to Golgi fragmentation mediated by non-muscle myosin IIA (NMIIA)/core2 N-acetylglucosaminyltransferase-M (C2GnT-M, GCNT3) complex formation[59]. Here, we were unable to establish viable HEK293 cells with KO of HSP90. Reduction of NSC80997 to NSC80997-KR decreased the Tn induction effect twofold (Fig. 3e, f). Further dissection of these compounds is needed to more specifically reveal the functional group(s) that mediate the Golgi fragmentation and abrogation of glycan elaboration.

We ruled out that NSC80997 and NSC255112 act like BFA[91]. First of all, these compounds do not impair protein secretion (Supplementary Fig. 6C). Secondly, because they induce fragmentation of the Golgi apparatus rather than the redistribution of Golgi into the ER induced by BFA (Fig. 5e, g and Supplementary Fig. 6)[92]. Interestingly, this process involves the initial formation of short tubules that later fragment into smaller structures and disperse throughout the cytoplasm[62]. Thirdly, the compounds do not affect the recruitment of the coatomer component β-COP, and this is a key step for the initiation of the BFA-dependent tubulation[93]. The lack of effect on the microtubules rules out that the fragmentation is the result of the well-established effect of microtubule-disrupting agents on the Golgi apparatus[94]. Therefore, the effect of the NSC80997 and NSC255112 compounds on the Golgi structure is mediated by a completely different mechanism(s), which will require extensive efforts to be fully dissected.

The finding that NSC80997 and NSC255112 rapidly and reversibly disrupts cellular glycosylation without substantially altering cell viability and capacity for secretion of recombinant glycoproteins makes these compounds highly attractive for use to probe whether proteins are glycosylated and explore functions of glycoproteins at the cell surface. To illustrate this, we demonstrated applications related to cancer immunotherapy and viral infections. Thus, pulsing target cells with the inhibitor compounds induced the truncated cancer-associated O-glycan Tn[29,63], which triggered ADCC activity by mAbs directed to Tn-glycopeptide epitopes in MUC1 and FXYD5 (Fig. 6c). Such antibodies are promising for therapeutic interventions[58,84] and appropriate cell lines to study their ADCC activity are missing[66]. We also demonstrated the use of the inhibitor compounds to explore the effect of glycosylation on SARS-CoV-2 infection, validating the importance of HS in the binding and infection (Fig. 7). The NSC80997 and NSC255112 compounds may become useful reagents to reversibly interfere with Golgi glycosylation for probing the contribution of elaborated glycans in biological interactions.

In summary, we developed an elaborate cell-based assay for screening of GalNAc-T isoenzymes inhibitors, and although our initial screening of a small compound library did not identify specific inhibitors the assay was validated and can be used for further screening of larger compound libraries. We did identify two potent reversible inhibitors that selectively affect Golgi-located glycosylation without affecting viability and secretion, and these compounds should be widely applicable and simple tools to investigate the role of glycosylation.

## Methods

### Small-molecule library and reagents

The small-molecule compound library was obtained from the National Cancer Institute (NCI), Division of Cancer Treatment and Diagnosis (DCTD), Developmental Therapeutics Program (DTP) (http://dtp.cancer.gov), and comprises the *Diversity Set V*, *Approved Oncology Drug Set VIII*, and the *Natural Product Set IV*. Compounds were provided dissolved in DMSO at a concentration of 10 mM and were diluted further in DMSO to a working concentration of 2 mM prior to use. The procedure for ketone reduction of NSC80997 is described in the supplementary information.

### Cell culture

HEK293 cells (#85120602, Sigma), HeLa (CCL-2, ATCC) and AGS cells (CRL-1739, ATCC) were maintained in DMEM (Sigma) supplemented with 10% fetal bovine serum (Sigma) and 4 mM GlutaMAX (Gibco). MCF7 (HTB-22, ATCC) were cultured in DMEM low glucose (Gibco) containing 10% fetal bovine serum, 4 mM GlutaMAX and 10 μg/mL insulin (Gibco). SH-SY5Y (CRL-2266, ATCC) cells were maintained in 50% RPMI-1640 medium (Sigma), 50% DMEM low glucose, supplemented with 10% fetal bovine serum and 4 mM GlutaMAX. Suspension HEK293-6E cells[95] were grown in serum-free Freestyle F17 (Gibco) supplemented with 0.1% Kolliphor P188 (Sigma) and 4 mM GlutaMax under constant agitation (120 rpm). CHOZN GS−/− cells (Sigma) were grown in 50% BalanCD CHO growth medium (Irvine Scientific) and 50% Ex-Cell CD medium (Sigma) supplemented with 8 mM GlutaMAX. All cells were cultured in humidified incubators at 37 °C and 5% $CO_2$.

### Gene targeting in HEK293 cells

CRISPR/Cas9 KO of genes was performed as previously described[96]. The guide RNAs targeting for human glucocorticoid receptor (GR) and androgen receptor (AR) are listed in Supplementary Table 1 and gRNAs and ZFNs for all glycoengineering performed were reported previously[64,97,98]. Briefly, HEK293 cells grown in 6-well plates (60-80% confluency) were co-transfected with 1 μg of the respective gRNA and 1 μg GFP-tagged Cas9-PBKS using Lipofectamine 3000 (Invitrogen) following the manufacturer's instructions. 24 h post-transfection, GFP-positive cells were bulk sorted and expanded for 7–8 days followed by single-cell sorting into 96-well plates using a cell sorter (SONY SH800). KO clones were selected by Indel Detection using Indel detection by amplicon analysis (IDAA)[96] and final clones were further verified by Sanger sequencing, and indel sequences of used clones are listed in Supplementary Table 1.

### Stable expression of N-, O-, and GAG glycoprotein reporters

HEK293 cells stably expressing this LDLR reporter were generated by targeted KI into the AAVS1 safe harbor site using a modified ObLiGaRe gene knock-in strategy as previously described[39,75]. Briefly, HEK cells growing in 6-well plates (60-80% confluency) were transfected with 1.5 μg LDLR reporter construct, and with 0.5 μg GFP-/E2 Crimson-tagged ZFN using Lipofectamine 3000. GFP/E2 Crimson double-positive cells were bulk sorted 24 h after transfection and single-cell sorted 7–8 days later. The clones were screened for correct integration by PCR using a primer pair specific for the junction area. Secreted and transmembrane MUC1 reporters with GFP-tag covering the densely O-glycosylated tandem repeat region of MUC1 were previously reported[39,43]. HEK293[WT], HEK293[KO COSMC], and HEK293[KO GCNT1/ST3GAL1/2] cells stably expressing the MUC1 reporters were selected by 2 weeks culture in the presence of 0.32 μg/mL G418 followed by 2 rounds of FACS enrichment for GFP expression. The secreted LDLR reporter construct was generated by fusion of 4 repeats of class A domain (A1−A4, aa1−1967, Uniprot P01130) of human LDLR with myc-, 6xHis, and ECFP tags. An expression construct containing the entire coding sequence of human erythropoietin (EPO) and the extracellular domain of the human proteoglycan serglycin fused with human Fc were

transfected into CHOZN GS−/− cells and high expressing clones were selected by glutamine selection[52,75].

### Compound library screen

HEK293[KO COSMC] and HEK293[KO GCNT1/ST3GAL1/2] cells stably secreting the LDLR or the MUC1 reporter, respectively, were seeded into round bottom 96-well plates (Nunc) at a density of $1 \times 10^5$ cells in 200 μl culture media containing 10 μM final concentration of screening compound, and an equal concentration of DMSO was used as control on every plate. The plates were incubated for 24 h and supernatants and cells were collected separately for enzyme-linked lectin assay (ELLA) and MTT assays, respectively. For ELLA, MaxiSorp 96-well plates (Nunc) were pre-coated with 1 μg/mL anti-6xHis antibody (R&D Systems) diluted in carbonate-bicarbonate buffer (pH 9.6) overnight at 4 °C followed by blocking with PLI-P buffer (0.5 M NaCl, 30 mM KCl, 0.0015 M $KH_2PO_4$, 0.0065 M $NaHPO_4$, 1% Triton X-100, 1% BSA, pH 7.4) for 2 h at RT. After washing with PBS-T (1x PBS, 0.05% Tween-20), supernatant was added, and plates were incubated overnight at 4 °C. Following extensive washing, plates were incubated with 2 μg/mL biotinylated VVA Lectin (Vector Laboratories) for 1 h at RT, washed, and incubated with 2 μg/mL streptavidin-HRP conjugate (Invitrogen) for 1 h at RT. Plates were extensively washed and developed by adding TMB substrate solution (Thermo Fisher). The reaction was stopped by adding 0.2 M $H_2SO_4$ and absorbance was measured at 450 nm using a HTX microplate reader (BioTek). For the MTT assays, cells were washed once with 1x PBS and resuspended in 70 μl pre-warmed culture media containing 0.5 mg/mL MTT (3-(4,5-dimethylthiazol-2-yl)−2,5-diphenyltetrazolium bromide) (Invitrogen). Following 30 min incubation at 37 °C, the cells were pelleted by centrifugation and dissolved in lysis buffer (isopropanol 90% (v/v), SDS 0.25% (w/v), 1 N HCL) and absorbance was measured at 595 nm using a HTX microplate reader.

### SDS-PAGE and western blot analysis

Supernatants containing the secreted reporters treated with NSC80997, NSC255112, Brefeldin-A (Invitrogen), or DMSO control were analyzed using NuPAGE Novex Bis-Tris 4 −12% gels (Invitrogen). The serglycin GAG reporter was visualized by staining gels with InstantBlue Coomassie protein stain (Expedeon). LDLR, MUC1 and EPO reporters were transferred to polyvinylidene difluoride (PVDF) membranes (60 min, 320 mA). The membranes were blocked in 5% skimmed milk in 1x PBS and washed three times in PBS-T prior to overnight incubation at 4 °C with mouse anti-6xHis antibody (R&D Systems) (1:1000) or Biotinylated VVA lectin (Vector lab) (0.2 μg/mL). After extensive washing with PBS-T, membranes were incubated with 1 μg/mL goat anti-mouse antibody Alexa Fluor 647 (Invitrogen) or 1 μg/mL streptavidin Alexa Fluor 647 conjugate (Invitrogen) and visualized using an ImageQuant LAS 4000 system (GE Healthcare).

### Flow cytometry analysis

Cells treated with compounds or DMSO control were stained with biotinylated lectins (Vector Laboratories) (VVA, PNA 0.2 μg/mL; Pan-Lectenz, 2,3-Lectenz, GSL II, PHA-L, GNL, DSL, RCAI, ECL, LCA, PHA-E, WGA 1 μg/mL) or mouse mAbs to Tn (1E3)[44,99], Tn-MUC1 (5E5)[65], FXYD5 (6C5 and NCC-MC53)[64] diluted 1:5000 in PBA (PBS with 1% (w/v) BSA) for 1 h at 4 °C. For SARS-CoV-2 spike protein binding, NSC80997, DMSO control or Heparinase mix (2.5 mU/mL HSase II, and 5 mU/mL HSase III; IBEX) treated cells were incubated with recombinant SARS-CoV-2 biotinylated spike protein S1/S2 (20 μg/mL) for 30 min at 4 °C. Spike protein was produced and biotinylated as previously described[71]. Cells were washed with PBA and incubated with streptavidin conjugated to Alexa Fluor 488 or 647 (Invitrogen), FITC-conjugated rabbit anti-mouse immunoglobulins (Dako), or Goat anti-mouse IgG, Alexa Flour 488 (Invitrogen) diluted 1:2000 in PBA, respectively. Cells were washed twice and resuspended in PBA and analyzed using a SA3800 spectral analyzer running the SA3800 software (SONY) or a

FACSCalibur instrument running BDFACStation software (BD Bioscience). Cells were gated to exclude dead cells and doublets (Supplementary Fig. 4B) and data was analyzed using FlowJo software (FlowJo, LCC).

## In vitro glycosylation assay

In vitro glycosylation assays were performed as product development assays in 80 μL buffer (25 mM cacodylic acid sodium, pH 7.4, 10 mM MnCl2, 0.25% Triton X-100), 250 μM UDP-GalNAc (Sigma), 0.45 μg/μL of purified LA1-4 and 400 nM of purified GalNAc-T11 at 37 °C. For time-course evaluation 5 μL of the reaction mixtures were taken at 0 min, 15 min, 30 min and 60 min, and quenched in 45 μL 0.1% TFA and diluted to 0.1 μM for matrix-assisted laser desorption ionization-time of flight (MALDI-TOF) analysis. Matrix-assisted laser desorption ionization-time of flight mass spectrometry (MALDI-TOF-MS) was performed in linear positive mode on a Bruker Autoflex instrument (Bruker Daltonik GmbH, Bremen, Germany) by mixing the quenched aliquots with a saturated solution of α-Cyano-4-hydroxycinnamic acid in ACN/H2O/TFA (70:30:0.1) at a 1:1 ratio on a target steel plate.

## Production and purification of reporter proteins

HEK or CHO cells stably expressing secreted MUC1 tandem repeat, EPO or serglycin reporters were seeded at a density of $0.5 \times 10^6$ cells/mL and cultured for 4 days in suspension. A secreted 6xHis-tagged E-cadherin reporter construct was transiently expressed HEK293-6E cells using Lipofectamine 3000 (Invitrogen). 24 h post-transfection cells were washed twice with PBS and incubated in fresh media containing 10 μM NSC80997 or DMSO and incubated for 48 h. For purification, cells were pelleted by centrifugation and supernatant was filtered through a 0.45 μm filter (VWR). The filtered supernatant was adjusted to a final concentration of 50 mM $NaHPO_4$, 300 mM NaCl and 10 mM imidazole and then run through a 200 μl Ni-NTA agarose column (Qiagen), washed with 20 c.v. of 50 mM $NaHPO_4$, 300 mM NaCl and 20 mM imidazole and eluted in 500 μl 50 mM $NaHPO_4$, 300 mM NaCl and 250 mM imidazole in 5 fractions. Fractions were checked by SDS-PAGE and positive fractions were pooled, desalted using Zeba™ spin desalting columns (Thermo Fisher) and quantified by Pierce™ BCA protein assay (Thermo Fisher) using the manufacturer's protocol.

## N-glycan analysis by MALDI-TOF

N-glycan analysis of the EPO reporter expressed from HEK293[WT] treated with NSC80997 or DMSO was performed as described previously[100]. Briefly, 10 μg purified EPO reporter was dried, resuspended in 10 μl 1% SDS in PBS and incubated at 60 °C for 30 min. Then 10 μl 2% NP-40/PBS containing 0.5 U PNGaseF (Roche) was added, and samples were incubated at 37°C overnight. The ethyl esterification derivatization was performed by adding 100 μL 0.25 M EDC/0.25 M HOBt in ethanol to the released glycan sample and incubating the mixture for 1 h at 37 °C. Cotton HILIC microtip purification was performed to obtain the purified and enriched N-glycans, and 3 μl of the mixture was spotted together with 1 μl super-DHB matrix (Sigma-Aldrich) onto a target steel plate. MALDI-TOF-MS spectra were recorded in reflectron positive mode on a Bruker Autoflex instrument (Bruker Daltonics) Spectra were recorded with an $m/z$-range of 1000–5000 Da.

## E-cadherin O-mannosylation analysis

Ten micrograms of purified E-cadherin reporter expressed in HEK293[WT] treated with 10 μM NSC80997 or DMSO was run on an SDS-PAGE and visualized by Coomassie stain. The E-cadherin band was cut-out followed by 3 times destaining in 100 mM Ambic buffer with 40% acetonitrile (ACN; Thermo Fisher) and dehydrated in 100% ACN. Dehydrated gel pieces were rehydrated in 50 μl 100 mM Ambic containing 200 ng Trypsin (Roche) and incubated at 37°C overnight. Digested peptides were collected by addition of 50 μl 50% ACN and

desalted using C18 stage tip. Peptides from control and NSC80997 treated samples were differentially labeled by a TMT6plex kit (Thermo Fisher) before being mixed, desalted using C18 stage tip and resuspended in 0.1% formic acid for liquid chromatography–mass spectrometry (LC-MS) analysis.

EASY-nLC 1200 UHPLC (Thermo Scientific) was interfaced via a PicoView nanoSpray ion source (New Objectives) to an Orbitrap Lumos mass spectrometer (Thermo Scientific). nLC was operated in a single analytical column setup using PicoFrit Emitters (New Objectives, 75-μm inner diameter) packed in-house with Reprosil-Pure-AQ C18 phase (Dr. Maisch, 1.9-μm particle size, ⌒19-cm column length), with a flow rate of 200 nl/min. Sample dissolved in 0.1% formic acid was injected onto the column and eluted in a gradient from 3 to 32% solvent B in 65 min, from 32 to 100% solvent B in 10 min, followed by isocratic elution at 100% solvent B for 15 min (total elution time 90 min, solvent A: 0.1%FA, solvent B: 80% acetonitrile in 0.1% FA). The nanoSpray ion source was operated at 2.1-kV spray voltage and 300 °C heated capillary temperature. A precursor MS1 scan ($m/z$ 350–2000) of intact peptides was acquired in the Orbitrap at a nominal resolution setting of 120,000. The five most abundant multiply charged precursor ions in the MS1 spectrum at a minimum MS1 signal threshold of 50,000 were triggered for sequential Orbitrap HCD MS2 and ETD MS2 ($m/z$ of 100–2000). MS2 spectra were acquired at a resolution of 50,000 for HCD MS2 and 60,000 for ETD MS2. Activation times were 30 and 200 ms for HCD and ETD fragmentation, respectively; isolation width was 1.6 mass units for HCD MS2 and 3 mass units for ETD MS", and 1 microscan was collected for each spectrum. Automatic gain control targets were 1,000,000 ions for Orbitrap MS1 and 100,000 for MS2 scans, and the automatic gain control for the fluoranthene ion used for ETD was 300,000. Supplemental activation (30%) of the charge-reduced species was used in the ETD analysis to improve fragmentation. Dynamic exclusion for 60 s was used to prevent repeated analysis of the same components.

## Immunocytochemistry

HeLa or MCF-7 cells were seeded on glass coverslips in 24-well plates and allowed to adhere prior to treatment with NSC80997, NSC255112, Brefeldin-A (00-4506-51, Thermo Fischer Scientific), Nocodazole (1228, Tocris) or DMSO. After treatment, cells were fixed in 4% paraformaldehyde for 10 min and washed with 1x PBS. Extracellular staining was performed by incubating the samples with 1 μg/mL biotinylated VVA for 2 h followed by staining with 1 μg/mL streptavidin-Alexa Fluor 488 for 1 h. For intracellular staining, fixed cells were permeabilized by incubation in 1x PBS containing 0.2% (v/v) Triton X-100 for 10 min and incubated 1 h at RT with blocking buffer (1x PBS 5% (w/v) BSA). Samples were incubated with primary antibodies directed against the glucocorticoid receptor (PA1-510A, Thermo Fisher), Giantin (mouse: ab24586, Abcam; rabbit: ab80864, Abcam), bCOP (ab2899, Abcam), GM130 (610822, BD-Biosciences), Tubulin (T6199, Sigma), AcTub (T7451, Sigma) or TGN46 (ab50595, Abcam) diluted 1:300 in PBA or undiluted GALNT1 (4D8)[101] hybridoma supernatant for 1 h or 2 h for spinning-disk confocal microscopy and epifluorescence microscopy, respectively. This was followed by 1 h staining with respective CyTM3 AffiniPure Donkey Anti-Mouse IgG (ImmunoResearch), Goat anit-mouse IgG, Alexa Flour 647 (Invitrogen) or Goat anti-mouse IgG, Alexa Flour 488 (Invitrogen) antibodies. For metabolic labeling of sialoglycans, HeLa cells were incubated for 24 h with 100 μM Ac5SiaNPoc[57] in combination with or without NSC80997 and after fixation with 4% paraformaldehyde cells were permeabilized for 5 min using PBS containing 0.1% (w/v) saponin (Sigma). Subsequently, coverslips were incubated with blocking buffer (PBS, 0.1% saponin, 5% BSA) and washed thrice with PBS before 45 min reaction at 37 °C with click buffer (PBS, 0.1% (w/v) saponin, 500 μM CuSO4, 500 μM sodium ascorbate, 25 μM azide-PEG3-biotin (Sigma)). Cells were washed and incubated for 30 min with streptavidin Alexa Fluor 488 conjugate. All coverslips were

washed with PBS-BSA and PBS and mounted using Fluoromount-G mounting medium with or without DAPI (Thermo Fisher).

## Microscopy and image analysis

Epifluorescence Images were acquired using a fluorescence microscopy system (Leica) and analyzed using ImageJ (NIH). Spinning-disk confocal images were acquired using a Nikon Ti2 inverted fluorescence microscope equipped with Yokogawa spinning-disk confocal systems using a 20x objective for Golgi-morphology scoring or using a 60x objective as Z-stacks with 0.5 μm between frames for co-localization studies. The level of co-localization between two populations of proteins was assessed on Z-stacks using the Imaris 9.2.1(Bitplane) co-localization analysis plug-in. Results are presented as Pearson correlation coefficient, which represents the linear relationship of the signal intensity from the green and red channels of the analyzed image.

## Time-lapse imaging and microscopy

HeLa cells were seeded into 8-well μ-slide chambers (Ibidi) 24 h post-transfection with 0.5 μg pmScarlet-H_Giantin_C1 using Lipofectamine 3000. On the following day, cells were rinsed twice in Live Cell Imaging Solution (A1429DJ, Thermo Fisher Scientific) and equilibrated at 37 C in a humidified camber on Nikon Ti2 inverted fluorescence microscope. 10 μM BFA (00-4506-51, Thermo Fisher Scientific), 10 μM NSC80997, or DMSO was added to the chamber and time-lapse movies were collected as Z-stacks acquired at 1 stack pr minute with frames 0.5 μm apart using the 60x objective. pmScarlet-H_Giantin_C1 was a gift from Dorus Gadella (Addgene plasmid # 85049; http://n2t.net/addgene:85049).

## Differential proteomic analysis

HEK293[WT] cells treated for 24 h with 10 μM NSC80997 or DMSO control were resuspended in 0.1% RapiGest (Waters Corporation) and lysed by sonication. Lysates were heated for 10 min at 80 °C, followed by reduction (5 mM DTT, 60 °C, 30 min), alkylation (10 mM iodoacetamide, RT, 30 min), and digestion with trypsin (25 μg/sample) (Roche) at 37 °C overnight. Cleared, acidified digests were loaded onto equilibrated SepPak C18 cartridges (Waters) followed by washing with 0.1% TFA. Columns were washed further with 0.1% FA and peptides were eluted with 50% MeOH in 0.1% FA. Samples were prepared in triplicates: DMSO control $n = 3$, NSC80997 treated $n = 3$. Peptides were labeled using a TMTsixplex isobaric labeling kit (Thermo Fisher) and submitted to LC/MS analysis.

EASY-nLC 1000 UHPLC (Thermo Scientific) was interfaced via a PicoView nanoSpray ion source (New Objectives) to an Orbitrap Fusion mass spectrometer (Thermo Scientific). nLC was operated in a single analytical column setup using PicoFrit Emitters (New Objectives, 75-μm inner diameter) packed in-house with Reprosil-Pure-AQ C18 phase (Dr. Maisch, 1.9-μm particle size, ⌐19-cm column length), with a flow rate of 200 nl/min. Sample dissolved in 0.1% formic acid was injected onto the column and eluted in a gradient from 2 to 25% solvent B in 95 min, from 25 to 80% solvent B in 10 min, followed by isocratic elution at 80% solvent B for 15 min (total elution time 120 min, solvent A: 0.1%FA, solvent B: 100% acetonitrile in 0.1%FA). The nanoSpray ion source was operated at 2.1-kV spray voltage and 300 °C heated capillary temperature. A precursor MS1 scan (m/z 350–1700) of intact peptides was acquired in the Orbitrap at a nominal resolution setting of 120,000. Ten of the most abundant multiply charged precursor ions in the MS1 spectrum at a minimum MS1 signal threshold of 50,000 were triggered for Orbitrap HCD MS2 (m/z of 100–2000). MS2 spectra were acquired at a resolution of 50,000 for HCD MS2 with stepped (+/− 5) collision energy at 37%. Activation times were 30 ms for HCD fragmentation. Isolation width was 3 mass units, and 1 microscan was collected for each spectrum. Automatic gain control targets were 1,000,000 ions for Orbitrap MS1 and 100,000 for MS2 scans. Dynamic

exclusion for 60 s was used to prevent repeated analysis of the same components.

## Data analysis

MS data processing for all raw files was performed using Proteome Discoverer (PD) version 2.2 software (Thermo Fisher Scientific). For the differential proteomics data, raw files were searched with Sequest HT search engine against a concatenated human-specific database (Uni-Prot, March 2019, contacting 20,355 canonical entries). Trypsin digestion was restricted to full specificity, and a maximum of two missed cleavages. The precursor mass tolerance was set to 10 ppm and fragment ion mass tolerance to 0.02 Da. Carbamidomethylation on cysteine residues, TMT6plex on peptide N-Terminus and Lysine were used as a fixed modification. Methionine oxidation was used as variable modification, with a maximum of 10 variable modifications per peptide. The minimum peptide length was set to five amino acids. Peptide FDR level was set to 1% and minimum number of unique peptides for protein identification was set to 1. The Benjamini–Hochberg method was used to adjust p-values for multiple comparisons. The Reporter Ion Quantifier node was applied for TMT6plex quantification of reporter ions using total peptide level normalization. To compare the relative abundance of proteins between two conditions, the fold-change between the median of the replicates was calculated.

For the analysis of E-cadherin O-mannosylation, MS data processing of raw files was performed using Proteome Discoverer version 1.4. Raw files were searched with Sequest HT node allowing both full- and semispecific trypsin cleavage. Precursor mass tolerance was 10 ppm and fragment ion mass tolerance was 0.02 Da. Carbamidomethyl (cysteine) and TMT6plex was set as fixed modifications; TMT-129C and TMT-130N were used to label DMSO and NSC80997 treated samples, respectively. Variable modifications included methionine oxidation and hexose modification of serine and threonine residues. Glycopeptide identifications that passed 1% FDR were further validated by manual inspection of spectra to ensure the accuracy of the assignment.

## ADCC assay

HEK293[WT] control or membrane MUC1 reporter expressing cells were treated with 10 μM NSC80997 or DMSO for 4 h at 37 °C, washed with medium and re-seeded into 96-well flat bottom plates at a density of $5 \times 10^5$ cells in RPMI 1640 medium supplemented with 10% FBS and $1 \times 2$-mercaptoethanol (Gibco). Cells were allowed to adhere for 2 h and isotype, 1E3, 5E5, 6C5 or Mouse IgG isotype control (Invitrogen) mAbs (>2 μg/mL) were added 20 min prior to addition of peripheral blood lymphocytes (PBLs). Frozen PBLs from healthy donors were thawed and labeled with 5 μM CFSE (Invitrogen) according to the manufacturer's instructions and added at 5:1, 10:1, and 20:1 ratios to the HEK293 cells. After overnight incubation, all cells were harvested and labeled with viability dye and analyzed by flow cytometry. Percentage lysis was calculated normalized to respective isotype controls.

## Disaccharide analysis of CS/DS and HS GAGs

Analysis of disaccharide composition of cellular CS/DS and HS was performed largely as previously described[102]. In more detail, $10 \times 10^6$ HEK293 cells in suspension culture were treated with 10 μM NSC80997 or equal volume DMSO as control for 24 h before being washed in PBS and incubated overnight with 50 mM Tris-HCl (pH 7.6), 10 mM CaCl₂, 0.1% Triton X-100, and 1 mg/mL pronase (Roche) at 37 °C with constant rotation. Pronase was heat inactivated at 98 °C for 10 min, and 2 mM MgCl₂ and 250 U of benzonase (Sigma-Aldrich) was added to the samples followed by incubation at 37 °C for 2 h. Samples were acidified to pH 4–5 using acetic acid, before GAGs were extracted using DEAE (Sigma-Aldrich) columns (0.5 mL), equilibrated with 20 mM NaOAc, 100 mM NaCl, 0.1% Triton X-100 (pH 5.0). After loading the samples, columns were washed with equilibration buffer followed by 20 mM

NaOAc, 100 mM NaCl (pH 5.0), and elution of GAGs was done with 1.5 mL of 20 mM NaOAC, 1.25 M NaCl (pH 5.0). 4.5 mL of ice-cold 99.9% ethanol saturated with NaOAc was added to the elutes, and GAGs were pelleted and retrieved from the suspension by centrifugation and speed-vacuum dried. Samples were split in two, where half of the material was used for CS/DS analysis and half for HS analysis. CS/DS was digested using 10 mU chondroitinase ABC (Sigma-Aldrich) in 40 mM NaOAc, 1 µM CaCl$_2$, in a total volume of 40 µL, while HS was digested using 10 mU heparinases I, II, III in 40 mM NaOAc, 5 mM CaCl$_2$, in a total volume of 40 µL. After incubation at 37 °C overnight, digested GAGs were lyophilized and labeled with AMAC through addition of 5 µL 0.1 M AMAC in acetic acid/DMSO (vol/vol 3:17) to the samples followed by incubation at room temperature for 15 min. 5 µL of 1 M NaCNBH$_3$ was then added and mixtures were incubated at 45 °C for 3 h. Excess AMAC was removed by acetone precipitation performed 2x. AMAC-labeled disaccharide samples corresponding to $2 \times 10^6$ cells were dissolved in 2% acetonitrile and subsequently analyzed on a Waters Acquity UPLC system using a BEH C18 column (2.1 × 150 mm, 1.7 µm, Waters) detecting the fluorescence signal at 525 nm. For analysis of CS/DS 80 mM ammonium acetate (pH 5.5) was used as mobile phase A and for HS 150 mM ammonium acetate (pH 5.6) was used as mobile phase A, with acetonitrile used as mobile phase B for both CS/DS and HS. HPLC separation of disaccharides was performed using a flow rate of 0.2 mL/min and a gradient of mobile phase B increasing from 3 to 13% over 30 min. A mix of 20 pmol AMAC-labeled disaccharides standard (Iduron) was analyzed immediately prior to samples for identification and quantification of disaccharides in the samples.

### Preparation and infection by pseudotyped VSV
Vesicular Stomatitis Virus (VSV) pseudotyped with spike proteins of SARS-CoV-2 were generated according to a published protocol (Whitt, 2010). Briefly, HEK293T, transfected to express full-length SARS-CoV-2 spike proteins, were inoculated with VSV-G pseudotyped DG-luciferase (Kerafast, MA). After 2 h at 37 °C, the inoculum was removed and cells were refed with DMEM supplemented with 10% FBS, 50 U/mL penicillin, 50 mg/mL streptomycin, and VSV-G antibody (I1, mouse hybridoma supernatant from CRL-2700; ATCC). Pseudotyped particles were collected 20 h post-inoculation, centrifuged at 1320 x *g* to remove cell debris and stored at −80 °C until use.

Cells were seeded at 10,000 cells per well in a 96-well plate. The cells (60–70% confluence) were treated with increasing concentrations of NSC80997 for 24 h or HSases for 30 min at 37 °C in serum-free DMEM. Culture supernatant containing pseudovirus (20–100 mL) was adjusted to a total volume of 100 mL with PBS and the solution was added to the cells. After 4 h at 37 °C the media was changed to complete DMEM. Cells infected with Luciferase containing virus were analyzed by Bright-GloTM (Promega) using the manufacturers protocol. Briefly, 100 mL of luciferin lysis solution was added to the cells and incubated for 5 min at room temperature. The solution was transferred to a black 96-well plate and luminescence was detected using an EnSpire multimodal plate reader (Perkin Elmer).

### Infection by authentic SARS-CoV-2 virus
SARS-CoV-2 isolate USA-WA1/2020 (BEI Resources, #NR-52281) was propagated on Caco-2 cells and infectious units quantified by focus forming assay using TMPRSS2-Vero cells. Cells in 96-well plates were treated with increasing concentrations of NSC80997 for 24 h and infected with SARS-CoV-2 for 18 h at 37 °C in media containing dilutions of NSC80997. Multiplicities of infection (MOI) used were 1 for Caco-2 and Huh7.5 and 0.5 for TMPRSS2-Vero. The cells were fixed in 4% formaldehyde in PBS for 30 min at room temp and then washed with PBS. Cells were permeabilized with PBS, 1% BSA, 0.1% Triton-X-100 and stained with anti-Nucleocapsid antibody (GeneTex GTX135357)

followed by AlexaFluor 594 secondary antibody (Thermo Fisher Scientific) and Sytox Green (Thermo Fisher Scientific) as a nuclear counterstain. Images of 5 fields per well were acquired at 10x magnification in the Incucyte S3 (Sartorius), and images were analyzed for nuclei count and percent infection using the software tools onboard the Incucyte. Percent infection and cell viability were normalized to DMSO-treated infected wells. All work with SARS-CoV-2 was conducted in Biosafety Level-3 conditions at the University of California San Diego following the guidelines approved by the Institutional Biosafety Committee.

### Statistics and reproducibility
Statistical analysis was performed using GraphPad Prism version 8.4.2 (GraphPad Software). Two-tailed Student's *t*-test was used to calculate significance between two groups (*t*-test), and multiple comparisons were performed with one-way ANOVA followed by Dunnetts post hoc test (ANOVA).

All western blots and microscopy imaging experiments were repeated twice with similar results.

### Reporting summary
Further information on research design is available in the Nature Portfolio Reporting Summary linked to this article.

## Data availability
All data generated or analyzed during this study are included in this article and supplementary information files. The mass spectrometry proteomics data has been deposited to the ProteomeXchange Consortium via the PRIDE repository with identifier PXD029623. Source data are provided with this paper.

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

## Acknowledgements

This work was supported by the Lundbeck Foundation; the Novo Nordisk Foundation; the Danish National Research Foundation (DNRF107); the European Union's Horizon 2020 research and innovation program under the Marie Sklodowska-Curie grant agreement No 787684 (to C.B.); the Dutch Research Council VI.Veni.202.045 (to C.B.); a Career Award for Medical Scientists from the Burroughs Wellcome Fund and NIH grants K08AI130381 (to A.F.C.); The Carlsberg Foundation CF20-0412 (to R.L.M.); the EMBO fellowship ALTF 1553-2015 co-funded by the European Commission (LTFCOFUND2013, GA-2013-609409) and Marie Curie Actions (to E.L.N.); to ARAID, the Spanish Ministry of Science, Innovation and Universities (BFU2016-75633-P and PID2019-105451GB-I00) and Gobierno de Aragón (E34_R17 and LMP58_18) with FEDER (2014–2020) funds for "Building Europe from Aragón" for financial support (to R.H.-G.). The following reagent was deposited by the Centers for Disease Control and Prevention and obtained through BEI Resources, NIAID, NIH: SARS-Related Coronavirus 2, Isolate USA-WA1/2020, NR-52281. Illustrations were made using Biorender.com

## Author contributions

D.M.S., C.B., H.C., and Y.N. conceived and designed the study; E.L.N., R.K., T.D.M., R.L.M., S.Y.V., K.T.S., A.H., R.H-G., and R.W. contributed with experimental data and interpretation; T.M.C., A.F.G., A.E.C., X.Y., S.K.C., J.D.E., and A.F.C. contributed to the SARS-CoV-2 studies; J.F.A.P. and T.J.B. performed the ketone reduction study; D.M.S., C.B., H.C., and Y.N. wrote the manuscript, and all authors edited and approved the final version.

## Competing interests

University of Copenhagen has filed a patent application covering the design of the cell-based screening assay for inhibitors described here, which is currently pending. Application numbers are: EP3455635A1, PCT/EP2017/061385, USPTO 20190330601, the inventors are: Eric Bennett, Yoshiki Narimatsu (Y.N.), Catharina Steentoft, Zhang Yang, Ulla Mandel, and Henrik Clausen (H.C.). GlycoDisplay Aps, Copenhagen, Denmark, has obtained a license to the field of the patent application. Y.N. and H.C. are co-founders of GlycoDisplay Aps and hold ownerships in the company as well as served as unpaid consultants. The remaining authors declare no competing interests.
