## [Peer review file · Nature Communications]

Editorial Note: This paper was reviewed at a different Nature journal prior to transfer to Nature Communications. Any reports pertaining to the review process prior to transfer are not shown here.

REVIEWERS' COMMENTS

Reviewer #1 (Remarks to the Author):

The revised manuscript has some interesting new findings that clarify some of the points raised in my previous review. The new data do show that the NSC80997 does not inhibit secretion at low concentrations, although there is some inhibition of secretion at 5 μ M, which is likely enhanced above this. Yet, there is a clear alteration of glycosylation at a concentration as low as 0.5 μ M, where secretion is unaffected.

The mechanism of action is still unclear, and while there are clear differences between BFA's action and that of NSC80977 (not the least that BFA does inhibit secretion), the lack of mechanism does not allow us to conclude how the new inhibitor is different from BFA.

To note is the fact that the introduction and abstract still mention Hsp90 as a possible target, even though new experiments detailed both in the manuscript and the rebuttal do say that this target mechanism has now been ruled out.

Reviewer #2 (Remarks to the Author):

This manuscript describes a cell-based screen for inhibitors of O-GalNAc transferases using the generation of glycoengineered cells expressing tailored reporter glycoproteins. While not successful in identifying GalNAcT11-specific inhibitors, the authors found two compounds that broadly inhibit Golgi-based glycosylation processes. The authors claim that these compounds possibly cause Golgi fragmentation that is reversible with removal of the drug. They provide evidence for functional defects in glycosylation after treatment with one drug using two different experimental systems (SARS-CoV2 infection and ADCC). The cell-based screen is well designed and novel, as is the identification that these two NCS compounds affect glycosylation.

The data are clearly and effectively presented throughout the manuscript with helpful schematics to explain the experiments. The effects of these compounds on glycosylation are mostly robust and there are no concerns with statistical analyses or treatment of uncertainties.

The conclusions made regarding effects on Golgi glycosylation are valid and carefully considered. These conclusions are well supported by analytical methods in many cases that provide further insight on the basis for electrophoretic mobility shifts noted by Western blotting.

The authors provide sound control experiments throughout the manuscript that greatly aid in distinguishing the effects of these two drugs from other known molecules. This will help other researchers in choosing when to utilize such compounds for cell-based studies. In summary, this is an interesting and noteworthy advance in the field.

January 25, 2023

Point-by-point response to Reviewers comments

Referee #1:

Reviewer #1 (Remarks to the Author):

Query #1: To note is the fact that the introduction and abstract still mention Hsp90 as a possible target, even though new experiments detailed both in the manuscript and the rebuttal do say that this target mechanism has now been ruled out.

Response #1: We can agree and the abstract and introduction have been revised to remove mentioning of HSP90. It is worth noting though that the new data provided does not rule out HSP90 as being involved, but only demonstrates that the HSP90 mediated loss of tubulin acetylation is not the direct cause of the observed phenotype.

Action #1: The abstract and introduction have been revised to remove mentioning of HSP90.

Referee #2:

Reviewer #2 (Remarks to the Author):

No queries